# *Drosophila* PSI controls circadian period and the phase of circadian behavior under temperature cycle via *tim* splicing

Lauren E Foley[1], Jinli Ling[1], Radhika Joshi[1], Naveh Evantal[2], Sebastian Kadener[2,3], Patrick Emery[1]*

[1]Department of Neurobiology, University of Massachusetts Medical School, Worcester, United States; [2]Hebrew University of Jerusalem, Jerusalem, Israel; [3]Brandeis University, Waltham, United States

**Abstract** The *Drosophila* circadian pacemaker consists of transcriptional feedback loops subjected to post-transcriptional and post-translational regulation. While post-translational regulatory mechanisms have been studied in detail, much less is known about circadian post-transcriptional control. Thus, we targeted 364 RNA binding and RNA associated proteins with RNA interference. Among the 43 hits we identified was the alternative splicing regulator P-element somatic inhibitor (PSI). PSI regulates the thermosensitive alternative splicing of *timeless* (*tim*), promoting splicing events favored at warm temperature over those increased at cold temperature. *Psi* downregulation shortens the period of circadian rhythms and advances the phase of circadian behavior under temperature cycle. Interestingly, both phenotypes were suppressed in flies that could produce TIM proteins only from a transgene that cannot form the thermosensitive splicing isoforms. Therefore, we conclude that PSI regulates the period of *Drosophila* circadian rhythms and circadian behavior phase during temperature cycling through its modulation of the *tim* splicing pattern.

*For correspondence:
Patrick.Emery@umassmed.edu

**Competing interests:** The authors declare that no competing interests exist.

## Introduction

Circadian rhythms are the organism's physiological and behavioral strategies for coping with daily oscillations in environment conditions. Inputs such as light and temperature feed into a molecular clock via anatomical and molecular input pathways and reset it every day. Light is the dominant cue for entraining the molecular clock, but temperature is also a pervasive resetting signal in natural environments. Paradoxically, clocks must be semi-resistant to temperature: they should not hasten in warm summer months or lag in the winter cold (this is called temperature compensation), but they can synchronize to the daily rise and fall of temperature (temperature entrainment) (*Pittendrigh, 1960*). Not only can temperature entrain the clock, it also has a role in seasonal adaptation by affecting the phase of behavior (see for example *Majercak et al., 1999*).

Molecular circadian clocks in eukaryotes are made up of negative transcriptional feedback loops (*Dunlap, 1999*). In *Drosophila,* the transcription factors CLOCK (CLK) and CYCLE (CYC) bind to E-boxes in the promoters of the clock genes *period (per)* and *timeless (tim)* and activate their transcription. PER and TIM proteins accumulate in the cytoplasm where they heterodimerize and enter the nucleus to feedback and repress the activity of CLK and CYC and thus downregulate their own transcription (*Hardin, 2011*). This main loop is strengthened by a scaffolding of interlocked feedback loops involving the transcription factors *vrille* (*vri*), *PAR domain protein 1* (*Pdp1*) and *clockwork orange* (*cwo*). Post-translational modifications are well-established mechanisms for adjusting the speed and timing of the clock (*Tataroglu and Emery, 2015*).

Increasing evidence indicates that post-transcriptional mechanisms controlling gene expression are also critical for the proper function of circadian clocks in many organisms. In *Drosophila*, the post-transcriptional regulation of *per* mRNA has been best studied. *per* mRNA stability changes as a function of time (*So and Rosbash, 1997*). In addition, *per* contains an intron in its 3'UTR (dmpi8) that is alternatively spliced depending on temperature and lighting conditions (*Majercak et al., 1999*; *Majercak et al., 2004*). On cold days, the spliced variant is favored, causing an advance in the accumulation of *per* transcript levels as well as an advance of the evening activity peak. This behavioral shift means that the fly is more active during the day when the temperature would be most tolerable in their natural environment. The temperature sensitivity of dmpi8 is due to the presence of weak non-canonical splice sites. However, the efficiency of the underlying baseline splicing is affected by four single nucleotide polymorphisms (SNPs) in the *per* 3'UTR that vary in natural populations and form two distinct haplotypes (*Low et al., 2012*; *Cao and Edery, 2017*). Also, while this splicing is temperature-sensitive in two *Drosophila* species that followed human migration, two species that remained in Africa lack temperature sensitivity of dmpi8 splicing, (*Low et al., 2008*). Furthermore, *Zhang et al. (2018)* recently demonstrated that the the *trans*-acting splicing factor B52 enhances dmpi8 splicing efficiency, and this effect is stronger with one of the two haplotypes. *per* is also regulated post-transcriptionally by the TWENTYFOUR-ATAXIN2 translational activation complex (*Zhang et al., 2013*; *Lim et al., 2011*; *Lim and Allada, 2013a*; *Lee et al., 2017*). This complex works by binding to *per* mRNA as well as the cap-binding complex and poly-A binding protein. This may enable more efficient translation by promoting circularization of the transcript. Interestingly, this mechanism appears to be required only in the circadian pacemaker neurons. Non-canonical translation initiation has also been implicated in the control of PER translation (*Bradley et al., 2012*). Regulation of PER protein translation has also been studied in mammals, with RBM4 being a critical regulator of mPER1 expression (*Kojima et al., 2007*). In flies however, the homolog of RBM4, LARK, regulates the translation of DBT, a PER kinase (*Huang et al., 2014*). miRNAs have emerged as important critical regulators of circadian rhythms in *Drosophila* and mammals, affecting the circadian pacemaker itself, as well as input and output pathways controlling rhythmic behavioral and physiological processes (*Tataroglu and Emery, 2015*; *Lim and Allada, 2013b*).

RNA-associated proteins (RAPs) include proteins that either bind directly or indirectly to RNAs. They mediate post-transcriptional regulation at every level. Many of these regulated events – including alternative splicing, splicing efficiency, mRNA stability, and translation – have been shown to function in molecular clocks. Thus, to obtain a broad view of the *Drosophila* circadian RAP landscape and its mechanism of action, we performed an RNAi screen targeting 364 of these proteins. This led us to discover a role for the splicing factor P-element somatic inhibitor (PSI) in regulating the pace of the molecular clock through alternative splicing of *tim*.

## Results

### An RNAi screen for RNA-associated proteins controlling circadian behavioral rhythms

Under constant darkness conditions (DD) flies have an intrinsic period length of about 24 hr. To identify novel genes that act at the post-transcriptional level to regulate circadian locomotor behavior, we screened 364 genes, which were annotated in either Flybase (FB2014_03, *Thurmond et al., 2019*) or the RNA Binding Protein Database (*Cook et al., 2011*) as RNA binding or involved in RNA associated processes, using period length as a readout of clock function (*Supplementary file 1*: RAP Screen Dataset). We avoided many, but not all, genes with broad effects on gene expression, such as those encoding essential splicing or translation factors. When possible, we used at least two non-overlapping RNAi lines from the TRiP and VDRC collections. RNAi lines were crossed to two different GAL4 drivers: *tim-GAL4* (*Kaneko et al., 2000*) and *Pdf-GAL4* (*Renn et al., 1999*) each combined with a *UAS-dicer-2* transgene to enhance the strength of the knockdown (*Dietzl et al., 2007*). These combinations will be abbreviated as *TD2* and *PD2*, respectively. *tim-GAL4* drives expression in all cells with circadian rhythms in the brain and body (*Kaneko et al., 2000*), while *Pdf-GAL4* drives expression in a small subset of clock neurons in the brain: the PDF-positive small (s) and large (l) LNvs (*Renn et al., 1999*). Among them, the sLNvs are critical pacemaker neurons that drive circadian behavior in DD (*Renn et al., 1999*; *Stoleru et al., 2005*). In the initial round of screening, we tested

the behavior of 4–8 males for each RNAi line crossed to both *TD2* and *PD2* (occasionally, fewer males were tested if a cross produced little progeny). We also crossed some RNAi lines to *w*[1118] (+) flies (most were lines selected for retest, see below). We noticed that *RNAi/+* control flies for the TRiP collection were 0.3 hr shorter than those of the VDRC collection (*Figure 1A*). Furthermore, the mean period from all RNAi lines crossed to either *PD2* or *TD2* was significantly shorter for the TRiP collection than for the VDRC collection (*Figure 1A*) (0.2 hr, *TD2* crosses; 0.5 hr, *PD2* crosses). We also found that many of the VDRC KK lines that resulted in long period phenotypes when crossed to both drivers contained insertions in the 40D locus (VDRC annotation), although this effect was stronger with *PD2* than *TD2*. It has been shown that this landing site is in the 5'UTR of *tiptop* (*tio)* and can lead to non-specific effects in combination with some GAL4 drivers, likely due to misexpression of *tio* (*Vissers et al., 2016*; *Green et al., 2014*). Indeed, when we crossed a control line that contains a UAS insertion at 40D (*40D-UAS*) to *PD2*, the progeny also had a ca. 0.6 hr longer period relative to the *PD2* control (*Figure 1B*). Thus, in order to determine a cutoff for candidates to further investigate, we analyzed the data obtained in our screen from the TRiP, VDRC, and the 40D KK VDRC lines independently (*Figure 1C*). These data are represented in two overlaid histograms that show period distributions: one for the *TD2* crosses (blue) and one for the *PD2* crosses (magenta). We chose a cutoff of two standard deviations (SD) from the mean period length for each RNAi line set. RNAi lines were selected for repeat if knockdown resulted in period lengths above or below the 2-SD cutoff. We also chose to repeat a subset of lines that did not pass the cutoff but were of interest and showed period lengthening or shortening, as well as lines that were highly arrhythmic in constant darkness (DD) or had an abnormal pattern of behavior in a light-dark cycle (LD). After a total of three independent experiments, we ended up with 43 candidates (*Table 1*) that passed the period length cutoffs determined by the initial screen; 31 showed a long period phenotype, while 12 had a short period. One line showed a short period phenotype with *PD2* but was long with *TD2* (although just below the 2-SD cutoff). Although loss of rhythmicity was also observed in many lines (*Supplementary file 1*), we decided to focus the present screen on period alterations to increase the probability of identifying proteins that regulate the circadian molecular pacemaker. Indeed, a change in the period length of circadian behavior is most likely caused by a defect in the molecular pacemaker of circadian neurons, while an increase in arrhythmicity can also originate from disruption of output pathways, abnormal development of the neuronal circuits underlying circadian behavioral rhythms, or cell death in the circadian neural network, for example.

Among the 43 candidate genes (*Tables 1* and *2*), we noticed a high proportion of genes involved or presumed to be involved in splicing (17), including five suspected or known to impact alternative splicing. Perhaps not surprisingly, several genes involved in snRNP assembly were identified in our screen. Their downregulation caused long period phenotypes. We also noticed the presence of four members of the CCR4-NOT complex, which can potentially regulate different steps of mRNA metabolism, including deadenylation, and thus mediate translational repression. Their downregulation mostly caused short period phenotypes and tended to result in high levels of arrhythmicity. *Rga* downregulation, however, resulted in a long period phenotype, suggesting multiple functions for the CCR4-NOT complex in the regulation of circadian rhythms. Interestingly, two genes implicated in mRNA decapping triggered by deadenylation, were also identified, with long periods observed when these genes were downregulated. Moreover, POP2, a CCR4-NOT component, was recently shown to regulate *tim* mRNA and protein levels (*Grima et al., 2019*). Another gene isolated in our screen, SMG5, was also recently found to impact circadian behavior (*Ri et al., 2019*). This validates our screen.

## Knockdown of *Psi* shortens the period of behavioral rhythms

A promising candidate to emerge from our screen was the alternative splicing regulator PSI (*Labourier et al., 2001*; *Siebel et al., 1992*). Knockdown of *Psi* with both *TD2* and *PD2* crossed to two non-overlapping RNAi lines from the VDRC collection (*GD14067* and *KK101882*) caused a significant period shortening, compared to the *TD2/+* and *PD2/+* controls (*Figure 2A–E*, *Table 3*), which the experimental flies need to be compared to since the GAL4 drivers in the *TD2* and *PD2* combination cause a previously reported dominant ca. 0.8 hr period lengthening (*Figure 2C*, left panel (*TD2/ + compared to w*[1118])*; *Kaneko et al., 2000*; *Renn et al., 1999*; *Zhang and Emery, 2013*; *Zhang et al., 2013*). Importantly, the RNAi lines did not cause period shortening on their own (*Figure 2C* left panel, *Table 3*). While most experiments were performed at 25°C, we noticed that at

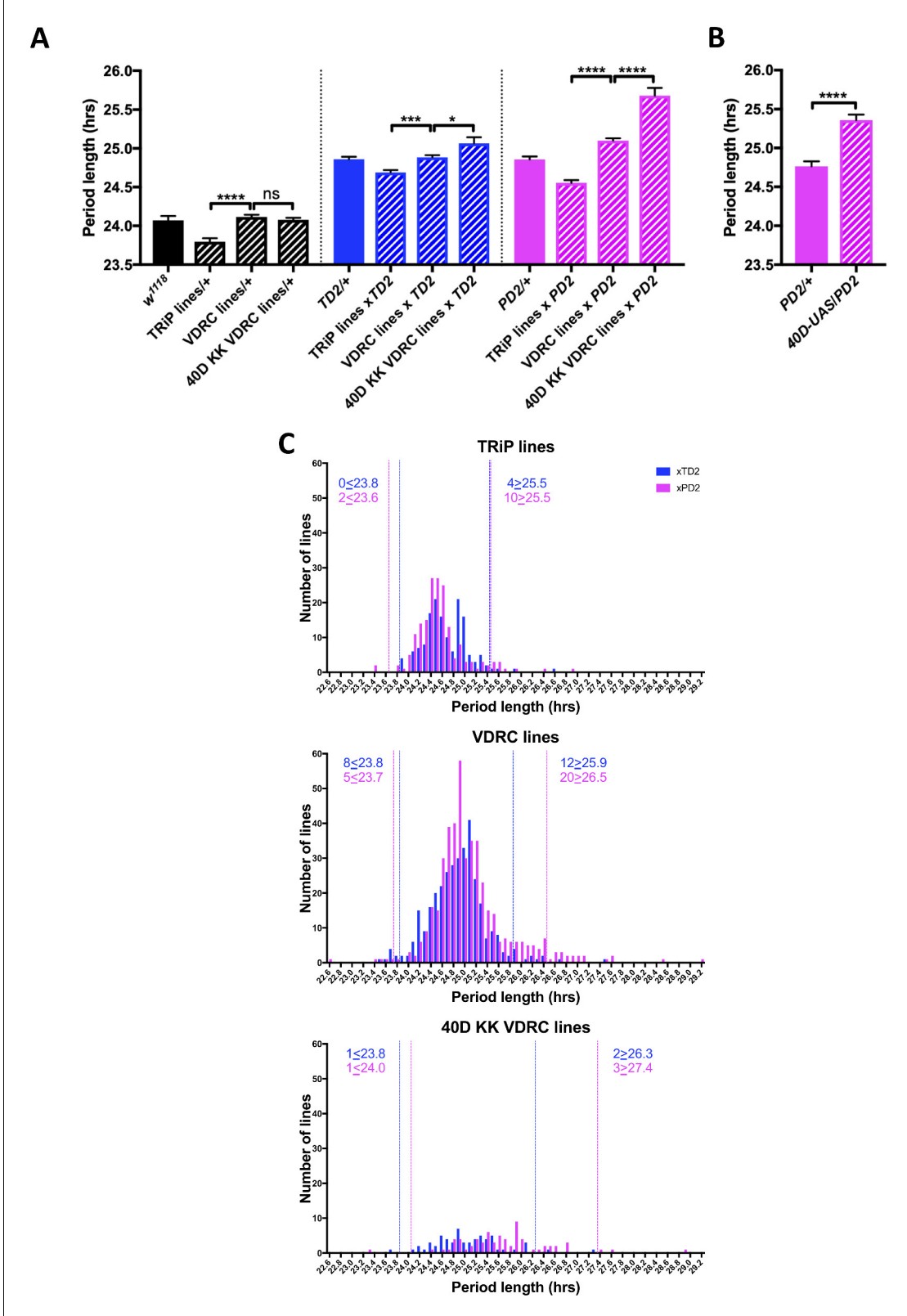

**Figure 1.** An RNAi screen of RNA associated proteins identifies long and short period hits. (A–B) Background effect of TRiP and VDRC collections on circadian period length. Circadian period length (hrs) is plotted on the y axis. RNAi collection and genotypes are labeled. Error bars represent SEM. (A) Left group (black bars): Patterned bars are the average of period lengths of a subset of RNAi lines in the screen crossed to $w^{1118}$ (TRiP/+ N = 17 crosses, VDRC/+ N = 46 crosses, 40D KK VDRC/+ N = 20 crosses). Solid bar is the $w^{1118}$ control (N = 20 crosses). Middle group (blue bars): Patterned
*Figure 1 continued on next page*

Figure 1 continued

bars are the average of period lengths of all RNAi lines in the screen crossed to *tim-GAL4, UAS-Dicer2 (TD2)* (*TRiP/TD2* N = 151 crosses, *VDRC/TD2* N = 340 crosses, *40D KK VDRC/TD2* N = 61 crosses). Solid bar is the *TD2/+* control (N = 35 crosses). Right group (magenta bars): Patterned bars are the average of period lengths of all RNAi lines in the screen crossed to *Pdf-GAL4, UAS-Dicer2 (PD2)* (*TRiP/PD2* N = 176 crosses, *VDRC/PD2* N = 448 crosses, *40D KK VDRC/PD2* N = 69 crosses). Solid bar is the *PD2/+* control (N = 36 crosses). One-way ANOVA followed by Tukey's multiple comparison test: *p<0.05, ***p<0.001, ****p<0.0001. Note that the overall period lengthening, relative to wild-type (*w[1118]*), when RNAi lines are crossed to *TD2* or *PD2* is a background effect of our drivers (see main text), while the period differences between the TRiP (shorter) and VDRC (longer) collections is most likely a background effect of the RNAi lines themselves. There is also a lengthening effect of the 40D insertion site in the VDRC KK collection that cannot be explained by a background effect, as it is not present in the RNAi controls (Left panel). Instead the lengthening was only observed when these lines were crossed to our drivers. A modest effect was seen with *TD2* (middle panel) and a larger effect was seen with *PD2* (right panel). (B) The period lengthening effect of the VDRC 40D KK lines is likely due to overexpression of *tio*, as we observed lengthening when a control line that lacks a RNAi transgene, but still has a UAS insertion in the 40D (*40D-UAS*) locus was crossed to *PD2*. N = 32 flies per genotype, ****p<0.0001, Unpaired Student's t-test. (C) Histogram of period lengths obtained in the initial round of screening. Number of lines per bin is on the y axis. Binned period length (hrs) is on the x axis. Bin size is 0.1 hr. *TD2* crosses are in blue and *PD2* crosses are in magenta. Dashed lines indicate our cutoff of 2 standard deviations from the mean. Number of crosses that fell above or below the cutoff is indicated. Top panel: TRiP lines. 0 lines crossed to *TD2* and 2 lines crossed to *PD2* gave rise to short periods and were selected for repeats. four lines crossed to *TD2* and 10 lines crossed to *PD2* gave rise to long periods and were selected for repeats. Middle panel: VDRC lines. eight lines crossed to *TD2* and 5 lines crossed to *PD2* gave rise to short periods and were selected for repeats. 12 lines crossed to *TD2* and 20 lines crossed to *PD2* gave rise to long periods and were selected for repeats. Bottom panel: VDRC 40D KK lines. one line crossed to *TD2* and 1 line crossed to *PD2* gave rise to short periods and were selected for repeats. two lines crossed to *TD2* and 3 lines crossed to *PD2* gave rise to long periods and were selected for repeats.

The online version of this article includes the following source data for figure 1:

**Source data 1.** 40D insertion control – behavior data.
**Source data 2.** Figure statistics – *Figure 1*.

30°C, *TD2/+* control had a period of ca. 24 hr (*Figure 2C*). We could thus meaningfully compare *TD2/RNAi* flies to both *RNAi/+* and *TD2/+* control at that temperature. The period of the experimental flies was significantly shorter than both controls (*Figure 2C*). Two additional lines from the TRiP collection (*JF01476* and *HMS00140)* also caused period shortening when crossed to *TD2* (*Table 1*). Interestingly, *HMS00140* targets only the *Psi-RA* isoform, indicating that the *RA* isoform is important for the control of circadian period (*Figure 2A*). Since four RNA lines caused a similar phenotype and only two of them partially overlapped (*Figure 2A*), we are confident that the period shortening was not caused by off-target effects. Moreover, both the *KK101882* and *GD14067* lines have been shown to efficiently downregulate *Psi* (*Guo et al., 2016*), and we confirmed by quantitative Real-Time PCR (qPCR) that the RNAi line *KK101882,* which gave the shortest period phenotype with *TD2,* significantly reduced *Psi* mRNA levels in heads (*Figure 2—figure supplement 1*). This line was selected for most of the experiments described below as it gave the strongest period phenotype.

The phenotype caused by *Psi* downregulation was more pronounced with *TD2* than with *PD2* (*Figure 2C–D*, *Table 3*). This was unexpected since the sLNvs - targeted quite specifically by *PD2* - determine circadian behavior period in DD (*Stoleru et al., 2005*; *Renn et al., 1999*). This could happen if *PD2* is less efficient at downregulating *Psi* in sLNvs than *TD2*, or if the short period phenotype is not solely caused by downregulation of *Psi* in the sLNvs. To distinguish between these two possibilities, we used *Pdf-GAL80* combined with *TD2* to inhibit *GAL4* activity specifically in the LNvs (*Stoleru et al., 2004*), while allowing RNAi expression in all other circadian tissues. With this combination, we also observed a significant period shortening compared to *TD2/+; Pdf-GAL80/+* controls, but the period shortening was not as pronounced as with *TD2* (*Figure 2E*, *Table 3*). We therefore conclude that both the sLNvs and non-PDF cells contribute to the short period phenotype caused by *Psi* downregulation (see discussion).

## *Psi* overexpression disrupts circadian behavior

Since we observed that downregulating *Psi* leads to a short period, we wondered whether overexpression would have an inverse effect and lengthen the period of circadian behavior. Indeed, when we overexpressed *Psi* by driving a *UAS-Psi* transgene (*Labourier et al., 2001*) with the *tim-GAL4* (*TG4*) driver, the period length of circadian behavior increased significantly by about 0.7 hr compared to the *TG4/+* control (*Figure 2F*, *Table 3*). Interestingly, we also observed a severe decrease

**Table 1.** Circadian behavior in DD of screen candidates

| Gene | RNAi Line | Driver | n | % of Rhythmic Flies | Period Average ±SEM | Power Average ±SEM |
|------|-----------|--------|---|---------------------|---------------------|--------------------|
| Atx-1 | GD11345 | TD2 | 24 | 75 | 26 ± 0.1 | 61.5 ± 4.1 |
| | | PD2 | 17 | 76 | 26.4 ± 0.1 | 50.7 ± 5.6 |
| | KK108861 | TD2 | 24 | 79 | 25.7 ± 0.1 | 49.1 ± 4.7 |
| | | PD2 | 23 | 74 | 26.2 ± 0.1 | 61.8 ± 4.5 |
| barc | GD9921 | PD2 | 20 | 75 | 26.5 ± 0.2 | 46.9 ± 5.6 |
| | KK101606** | TD2 | 6 | 83 | 25.3 ± 0.5 | 55.4 ± 12.7 |
| | | PD2 | 16 | 75 | 27 ± 0.4 | 43.9 ± 5.1 |
| bsf | JF01529 | TD2 | 24 | 88 | 25.8 ± 0.1 | 68.4 ± 4.6 |
| | | PD2 | 24 | 67 | 25.7 ± 0.1 | 47.6 ± 4.1 |
| CG16941 | GD9241 | PD2 | 8 | 0 | | |
| | HMS00157 | PD2 | 24 | 4 | 23.4 | 28.3 |
| | KK102272 | PD2 | 8 | 0 | | |
| CG32364 | HMS03012 | PD2 | 24 | 88 | 25.7 ± 0.1 | 58.9 ± 3 |
| CG42458 | KK106121 | TD2 | 23 | 35 | 26.5 ± 0.2 | 38.3 ± 4.9 |
| | | PD2 | 22 | 82 | 26.2 ± 0.1 | 71 ± 4.1 |
| CG4849 | KK101580 | TD2 | 1 | 0 | | |
| | | PD2 | 24 | 63 | 27.3 ± 0.2 | 48.8 ± 4.1 |
| CG5808 | KK102720* | TD2 | 23 | 70 | 27.4 ± 0.1 | 45.3 ± 5.1 |
| | | PD2 | 24 | 54 | 28.5 ± 0.6 | 34.8 ± 2.7 |
| CG6227 | GD11867 | TD2 | 1 | 0 | | |
| | | PD2 | 16 | 63 | 26.7 ± 0.2 | 51.4 ± 7 |
| | KK108174 | TD2 | 4 | 0 | | |
| | | PD2 | 20 | 30 | 24.2 ± 0.4 | 30.9 ± 3.5 |
| CG7903 | KK103182* | TD2 | 24 | 8 | 23.6 | 26.3 |
| | | PD2 | 24 | 75 | 26.4 ± 0.2 | 49.1 ± 3.7 |
| CG8273 | GD13870 | TD2 | 24 | 83 | 25.9 ± 0.1 | 47.3 ± 4.6 |
| | | PD2 | 14 | 100 | 25.4 ± 0.1 | 51.2 ± 4.8 |
| | KK102147 | TD2 | 24 | 58 | 25.5 ± 0.1 | 41.1 ± 5 |
| | | PD2 | 23 | 100 | 25.7 ± 0.1 | 64.3 ± 3.9 |
| CG8636 | GD13992 | PD2 | 12 | 50 | 26.9 ± 0.2 | 36 ± 6.4 |
| | KK110954 | TD2 | 1 | 0 | | |
| | | PD2 | 19 | 63 | 26.3 ± 0.3 | 51.4 ± 5.6 |
| CG9609 | HMS01000 | PD2 | 24 | 46 | 26.3 ± 0.2 | 46.1 ± 6.5 |
| | KK109846 | TD2 | 23 | 78 | 25.3 ± 0.1 | 48.5 ± 4.2 |
| | | PD2 | 23 | 91 | 26.3 ± 0.1 | 56.4 ± 3.9 |
| Cnot4 | JF03203 | TD2 | 23 | 26 | 23.7 ± 0.1 | 39.8 ± 6 |
| | | PD2 | 31 | 77 | 23.9 ± 0.1 | 51.1 ± 3.2 |
| | KK101997 | TD2 | 32 | 47 | 23.9 ± 0.1 | 37.3 ± 2.9 |
| | | PD2 | 27 | 93 | 25 ± 0.1 | 48 ± 4.1 |
| Dcp2 | KK101790 | TD2 | 22 | 64 | 26 ± 0.1 | 49.7 ± 5.3 |
| | | PD2 | 24 | 92 | 25.9 ± 0.1 | 62.5 ± 4.1 |
| eIF1 | KK109232* | PD2 | 24 | 4 | 23.2 | 68.9 |
| eIF3l | KK102071 | TD2 | 24 | 21 | 26 ± 0.2 | 28.9 ± 2.4 |
| | | PD2 | 23 | 100 | 25.7 ± 0.1 | 62.5 ± 3.9 |

*Table 1 continued on next page*

*Table 1 continued*

| Gene | RNAi Line | Driver | n | % of Rhythmic Flies | Period Average ±SEM | Power Average ±SEM |
|---|---|---|---|---|---|---|
| Hrb98DE | HMS00342 | PD2 | 22 | 91 | 25.8 ± 0.1 | 60.2 ± 4.1 |
| l(1)G0007 | GD8110 | PD2 | 24 | 63 | 26.3 ± 0.2 | 42.4 ± 3.7 |
| | KK102874 | TD2 | 24 | 17 | 26.9 ± 0.4 | 32.6 ± 5.5 |
| | | PD2 | 23 | 48 | 26.7 ± 0.2 | 48 ± 6.1 |
| LSm7 | GD7971 | PD2 | 22 | 36 | 28 ± 0.4 | 43.5 ± 5.6 |
| ncm | GD7819 | PD2 | 8 | 0 | | |
| | KK100829* | PD2 | 19 | 32 | 23.3 ± 0.1 | 34.4 ± 5.6 |
| Nelf-A | KK101005 | TD2 | 24 | 63 | 26.4 ± 0.1 | 52.9 ± 4.4 |
| | | PD2 | 23 | 74 | 24.8 ± 0.1 | 59.4 ± 4.5 |
| Not1 | GD9640 | PD2 | 23 | 4 | 22.6 | 43.6 |
| | KK100090 | PD2 | 10 | 30 | 23.8 ± 0.3 | 39.4 ± 4.7 |
| Not3 | GD4068 | PD2 | 8 | 0 | | |
| | KK102144 | PD2 | 21 | 14 | 23.6 ± 0.1 | 30.8 ± 2.1 |
| Patr-1 | KK104961* | TD2 | 23 | 30 | 26.3 ± 0.2 | 33.6 ± 3 |
| | | PD2 | 24 | 63 | 27.1 ± 0.2 | 38.3 ± 3.6 |
| Pcf11 | HMS00406 | PD2 | 8 | 13 | 24 | 20.1 |
| | KK100722 | PD2 | 24 | 21 | 23.3 ± 0.1 | 35.4 ± 5 |
| pcm | GD10926 | TD2 | 16 | 63 | 25.7 ± 0.1 | 36.6 ± 4.1 |
| | | PD2 | 20 | 55 | 26.3 ± 0.2 | 40.4 ± 3.8 |
| | KK108511 | TD2 | 24 | 21 | 25.7 ± 0.2 | 40.7 ± 7.8 |
| | | PD2 | 24 | 17 | 27.7 ± 0.6 | 32.9 ± 6.1 |
| Psi | GD14067 | TD2 | 48 | 79 | 23.7 ± 0.07 | 49.6 ± 3.0 |
| | | PD2 | 32 | 84 | 24.2 ± 0.1 | 53.3 ± 4.1 |
| | HMS00140 | TD2 | 24 | 100 | 24 ± 0.1 | 61.8 ± 4.2 |
| | | PD2 | 20 | 85 | 24.5 ± 0.1 | 52.9 ± 5.6 |
| | JF01476 | TD2 | 24 | 92 | 24 ± 0.1 | 64.7 ± 4.9 |
| | | PD2 | 24 | 92 | 24.3 ± 0.1 | 53.2 ± 4 |
| | KK101882 | TD2 | 35 | 77 | 23.6 ± 0.06 | 61.9 ± 3.7 |
| | | PD2 | 47 | 89 | 24.7 ± 0.06 | 56.3 ± 3.4 |
| Rga | GD9741 | TD2 | 24 | 21 | 26.2 ± 0.1 | 32.8 ± 3.2 |
| | | PD2 | 22 | 36 | 25.4 ± 0.2 | 36.1 ± 4.7 |
| RpS3 | GD4577 | PD2 | 14 | 57 | 26.4 ± 0.2 | 48.9 ± 5.9 |
| | JF01410 | PD2 | 24 | 50 | 25.6 ± 0.2 | 34.9 ± 2.3 |
| | KK109080 | PD2 | 8 | 38 | 26 ± 1.3 | 34.5 ± 6.3 |
| Rrp6 | GD12195 | PD2 | 10 | 10 | 24.5 | 27.2 |
| | KK100590 | PD2 | 21 | 10 | 23.6 | 43.2 |
| sbr | HMS02414 | TD2 | 13 | 85 | 26.8 ± 0.2 | 48.7 ± 5.3 |
| | | PD2 | 21 | 100 | 24.9 ± 0.1 | 57.2 ± 4.6 |
| Set1 | GD4398 | TD2 | 20 | 90 | 25.8 ± 0.1 | 52.1 ± 4.2 |
| | | PD2 | 13 | 77 | 25.3 ± 0.1 | 42.1 ± 5.5 |
| | HMS01837 | TD2 | 23 | 78 | 25.6 ± 0.1 | 47.9 ± 3.6 |
| | | PD2 | 24 | 92 | 24.8 ± 0.1 | 50 ± 3.8 |
| SmB | GD11620 | PD2 | 13 | 69 | 26.2 ± 0.1 | 52.1 ± 8 |

*Table 1 continued on next page*

Table 1 continued

| Gene | RNAi Line | Driver | n | % of Rhythmic Flies | Period Average ±SEM | Power Average ±SEM |
|---|---|---|---|---|---|---|
| | HM05097 | PD2 | 24 | 58 | 25.6 ± 0.1 | 45.2 ± 4.4 |
| | KK102021 | PD2 | 2 | 100 | 25.6 | 67.1 |
| SmE | GD13663 | PD2 | 24 | 58 | 25.7 ± 0.3 | 37.3 ± 3.3 |
| | HMS00074 | PD2 | 8 | 100 | 24.5 ± 0.1 | 55.1 ± 7.4 |
| | KK101450 | PD2 | 15 | 67 | 26.5± | 51.3 ± 7.8 |
| SmF | JF02276 | PD2 | 24 | 75 | 25.8 ± 0.1 | 46.3 ± 3.9 |
| | KK107814 | PD2 | 21 | 57 | 27.3 ± 0.3 | 45.4 ± 4.2 |
| smg | GD15460 | PD2 | 24 | 58 | 26.5 ± 0.2 | 39 ± 3.5 |
| Smg5 | KK102117 | TD2 | 23 | 52 | 23.7 ± 0.1 | 38.9 ± 3.7 |
| | | PD2 | 24 | 79 | 23.9 ± 0.1 | 58.5 ± 4.3 |
| Smn | JF02057 | TD2 | 3 | 67 | 24.2 | 25.9 |
| | | PD2 | 24 | 54 | 25.7 ± 0.1 | 47.2 ± 3.6 |
| | KK106152 | TD2 | 24 | 67 | 25.3 ± 0.1 | 39.7 ± 3.5 |
| | | PD2 | 24 | 96 | 26.3 ± 0.2 | 48.7 ± 2.7 |
| snRNP-U1-C | GD11660 | PD2 | 11 | 82 | 25.7 ± 0.1 | 56.5 ± 6.1 |
| | HMS00137 | PD2 | 24 | 92 | 25.8 ± 0.1 | 55.9 ± 4.1 |
| Spx | GD11072 | PD2 | 14 | 64 | 26.5 ± 0.2 | 56.1 ± 7.4 |
| | KK108243 | TD2 | 4 | 100 | 24 ± 0.2 | 47.5 ± 10.2 |
| | | PD2 | 19 | 79 | 26.9 ± 0.3 | 56.4 ± 5 |
| Srp54k | GD1542 | PD2 | 5 | 0 | | |
| | KK100462 | PD2 | 24 | 17 | 23.7 ± 0.4 | 31.3 ± 6 |
| Zn72D | GD11579 | TD2 | 28 | 89 | 26.3 ± 0.1 | 46.1 ± 4.6 |
| | | PD2 | 22 | 82 | 26.4 ± 0.1 | 59.4 ± 6.9 |
| | KK100696 | TD2 | 26 | 73 | 26.8 ± 0.1 | 57 ± 3.6 |
| | | PD2 | 24 | 83 | 26 ± 0.1 | 57 ± 4.5 |

*Line contains insertion at 40D.

** Unknown if line contains insertion at 40D.

in the number of rhythmic flies. When we overexpressed *Psi* with *Pdf-GAL4 (PG4)*, period was not statistically different from control (*PG4/+*), and rhythmicity was not reduced compared to the *UAS-Psi/+* control (*Figure 2G*). Overexpression of *Psi* with the *tim-GAL4; Pdf-GAL80* combination caused a severe decrease in rhythmicity but caused only a subtle period lengthening compared to *TG4/+; Pdf-GAL80/+* controls (*Figure 2H*, *Table 3*). The effect of *Psi* overexpression on period is in line with the knockdown results, indicating that PSI regulates circadian behavioral period through both PDF+ LNvs and non-PDF circadian neurons. However, the increase in arrhythmicity observed with *Psi* over-expression is primarily caused by non-PDF cells.

### *Psi* downregulation also shortens the period of body clocks

We wanted to further examine the effect of *Psi* knockdown on the molecular rhythms of two core clock genes: *period (per)* and *timeless (tim)*. To do this, we took advantage of two *luciferase* reporter transgenes. We downregulated *Psi* with the *TD2* driver in flies expressing either a TIM-LUCIFERASE (*ptim*-TIM-LUC) or a PER-LUCIFERASE (BG-LUC) fusion protein under the control of the *tim* or *per* promoter, respectively. We estimated period of luciferase activity rhythms over the first two days in DD, because oscillations rapidly dampened. Fully consistent with our behavioral results, the period of LUC activity was significantly shortened by about 1–1.5 hr compared to controls when *Psi* was downregulated in *ptim*-TIM-LUC flies (*Figure 2—figure supplement 2A and B*). Knockdown of *Psi* in

**Table 2.** Predicted or known functions of screen candidates

| Gene | Molecular function (based on information from Flybase) (*Thurmond et al., 2019*) |
|---|---|
| Atx-1 | RNA binding |
| barc | mRNA splicing; mRNA binding; U2 snRNP binding |
| bsf | mitochondrial mRNA polyadenylation, stability, transcription, translation; polycistronic mRNA processing; mRNA 3'-UTR binding |
| CG16941/Sf3a1 | alternative mRNA splicing; RNA binding |
| CG32364/tut | translation; RNA binding |
| CG42458 | mRNA binding |
| CG4849 | mRNA splicing; translational elongation |
| CG5808 | mRNA splicing; protein peptidyl-prolyl isomerization; regulation of phosphorylation of RNA polymerase II C-terminal domain; mRNA binding |
| CG6227 | alternative mRNA splicing; ATP-dependent RNA helicase activity |
| CG7903 | mRNA binding |
| CG8273/Son | mRNA processing; mRNA splicing; RNA binding |
| CG8636/eIF3g1 | translational initiation; mRNA binding |
| CG9609 | transcription; proximal promoter sequence-specific DNA binding |
| Cnot4 | CCR4-NOT complex |
| Dcp2 | deadenylation-dependent decapping of mRNA; cytoplasmic mRNA P-body assembly; RNA binding |
| eIF1 | ribosomal small subunit binding; RNA binding; translation initiation |
| eIF3l | translational initiation |
| Hrb98DE | translation; alternative mRNA splicing; mRNA binding |
| l(1)G0007 | alternative mRNA splicing; 3'—5' RNA helicase activity |
| LSm7 | mRNA splicing; mRNA catabolic process; RNA binding |
| ncm | mRNA splicing; RNA binding |
| Nelf-A | transcription elongation; RNA binding |
| Not1 | translation; poly(A)-specific ribonuclease activity; CCR4-NOT complex |
| Not3 | translation; transcription; poly(A)-specific ribonuclease activity; CCR4-NOT complex |
| Patr-1 | cytoplasmic mRNA P-body assembly; deadenylation-dependent decapping of mRNA; RNA binding |
| Pcf11 | mRNA polyadenylation; transcription termination; mRNA binding |
| pcm | cytoplasmic mRNA P-body assembly; 5'—3' exonuclease activity |
| Psi | alternative mRNA splicing; transcription; mRNA binding |
| Rga | translation; transcription; poly(A)-specific ribonuclease activity; CCR4-NOT complex |
| RpS3 | DNA repair; translation; RNA binding; structural constituent of ribosome |
| Rrp6 | chromosome segregation; mRNA polyadenylation; nuclear RNA surveillance; 3'—5' exonuclease activity |
| sbr | mRNA export from nucleus; mRNA polyadenylation; RNA binding |
| Set1 | histone methyltransferase activity; nucleic acid binding; contains an RNA Recognition Motif |
| SmB | mRNA splicing; RNA binding |
| SmE | mRNA splicing; spliceosomal snRNP assembly |
| SmF | mRNA splicing; spliceosomal snRNP assembly; RNA binding |
| smg | RNA localization; translation; mRNA poly(A) tail shortening; transcription; mRNA binding |

*Table 2 continued*

| Gene | Molecular function (based on information from Flybase) (*Thurmond et al., 2019*) |
|---|---|
| Smg5 | nonsense-mediated decay; ribonuclease activity |
| Smn | spliceosomal snRNP assembly; RNA binding |
| snRNP-U1-C | mRNA 5'-splice site recognition; mRNA splicing, alternative mRNA splicing |
| Spx | mRNA splicing; mRNA binding |
| Srp54k | SRP-dependent cotranslational protein targeting to membrane; 7S RNA binding |
| Zn72D | mRNA splicing; RNA binding |

BG-LUC flies resulted in a similar trend, although differences did not reach statistical significance (*Figure 2—figure supplement 2C and D*). Period was however shorter in experimental flies compared to both control genotypes in all four independent experiments performed with BG-LUC (and all six with *ptim*-TIM-LUC). Since the luciferase signal in these flies is dominated by light from the abdomen (*Lamba et al., 2018*; *Stanewsky et al., 1997*), this indicates that *Psi* knockdown, shortens the period of circadian clocks in peripheral tissues as well as in the brain neural network that controls circadian behavior.

## Alternative splicing of two clock genes, *cwo* and *tim*, is altered in *Psi* knockdown flies

PSI has been best characterized for its role in alternative splicing of the *P element transposase* gene in somatic cells (*Labourier et al., 2001*; *Siebel et al., 1992*). However, it was recently reported that PSI has a wider role in alternative splicing (*Wang et al., 2016*). Wang et al. reported an RNA-seq dataset of alternative splicing changes that occur when a lethal *Psi*-null allele is rescued with a copy of *Psi* in which the AB domain has been deleted (PSIΔAB). This domain is required for the interaction of PSI with the U1 snRNP, which is necessary for PSI to mediate alternative splicing of *P element transposase* (*Labourier et al., 2002*). Interestingly, *Wang et al. (2016)* found that PSIΔAB affects alternative splicing of genes involved in complex behaviors such as learning, memory and courtship. Intriguingly, we found four core clock genes listed in this dataset: *tim*, *cwo*, *sgg* and *Pdp1*. We decided to focus on *cwo* and *tim*, since only one specific splicing isoform of *Pdp1* is involved in the regulation circadian rhythm, (*Pdp1e*) (*Zheng et al., 2009*), and since the *sgg* gene produces a very complex set of alternative transcripts. After three days of LD entrainment, we collected RNA samples at four time points on the first day of DD and determined the relative expression of multiple isoforms of *cwo* and *tim* in *Psi* knockdown heads compared to driver and RNAi controls.

CWO is a basic helix-loop-helix (bHLH) transcriptional factor and is part of an interlocked feedback loop that reinforces the main loop by competing with CLK/CYC for E-box binding (*Matsumoto et al., 2007*; *Lim et al., 2007*; *Kadener et al., 2007*; *Richier et al., 2008*). There are three mRNA isoforms of *cwo* predicted in Flybase (*Figure 3—figure supplement 1A*) (*Thurmond et al., 2019*). Of the three, only *cwo-RA* encodes a full-length CWO protein. Exon two is skipped in *cwo-RB,* and in *cwo-RC* there is an alternative 3' splice site in the first intron that lengthens exon 2. Translation begins from a downstream start codon in *cwo-RB* and *-RC*, because exon two skipping or lengthening, respectively, causes a frameshift after the start codon used in *cwo-RA*. The predicted start codon in both *cwo-RB* and *cwo-RC* would produce an N-terminal truncation of the protein, which would thus be missing the basic region of the bHLH domain and should not be able to bind DNA. The *cwo-RB* and cwo-RC isoforms may therefore encode endogenous dominant negatives.

We found that the level of the *cwo-RB* isoform was significantly reduced compared to both controls at CT 9 (*Figure 3—figure supplement 1C*). The *cwo-RA* isoform was also reduced compared to both controls at CT9 (*Figure 3—figure supplement 1B*). This reduction was significant compared to the *TD2/+* control (p=0.0002) but was just above the significance threshold compared to the *PsiRNAiKK/+* control (p=0.0715). Conversely, *cwo-RC* isoform expression was significantly increased at CT 15 (*Figure 3—figure supplement 1D*). The overall expression of all *cwo* mRNAs in *Psi*

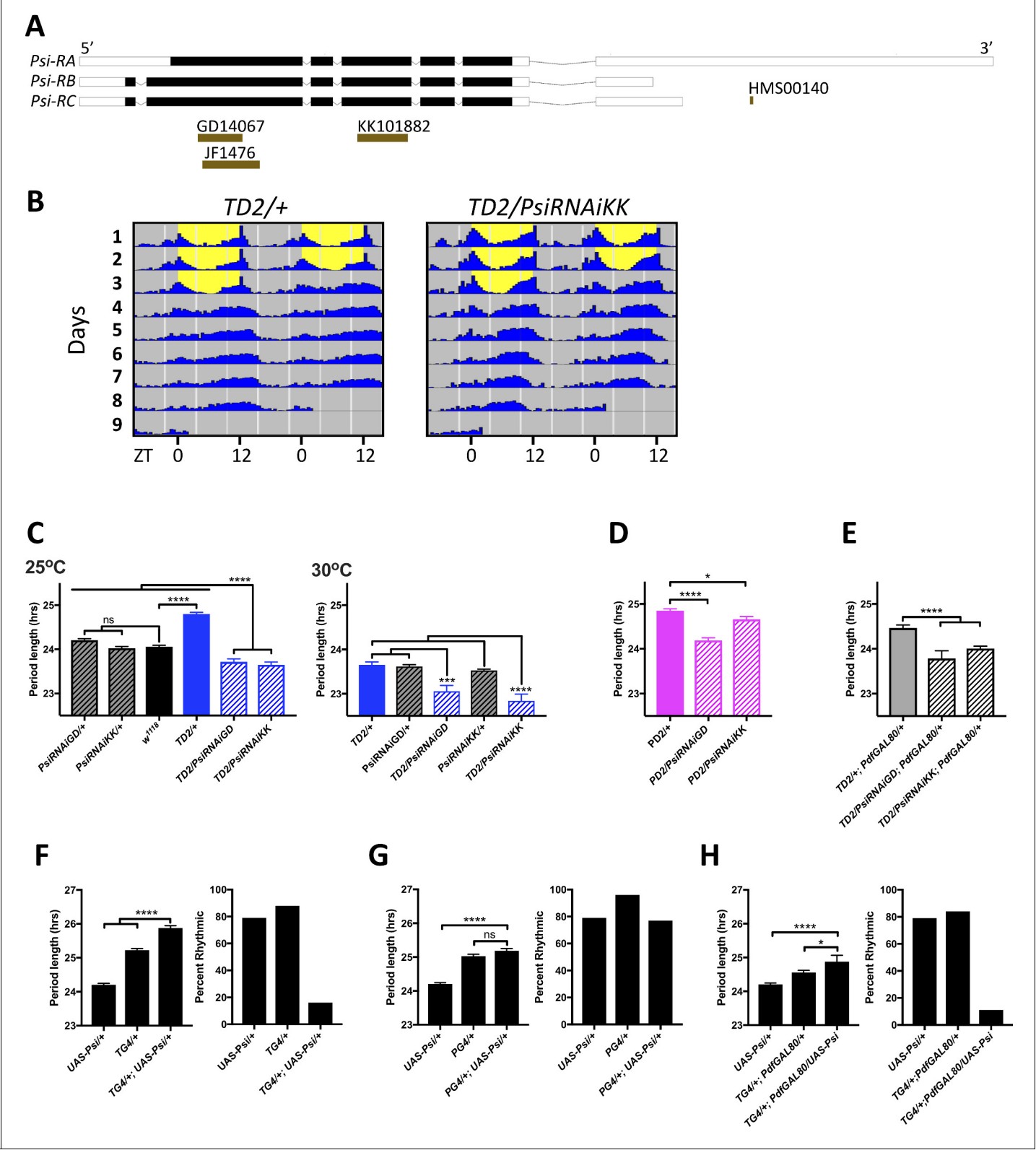

**Figure 2.** Expression level of *Psi* affects the circadian behavior period length and circadian rhythmicity. (**A**) Schematic of *Psi* isoforms and position of the long and short hairpins used in this study. Adapted from Ensembl 94 (*Zerbino et al., 2018*). (**B–E**) Knockdown of *Psi* shortens the behavioral period. (**B**) Double-plotted actograms showing the average activities during 3 days in LD and 5 days in DD. Left panel: *TD2/+* (control) flies. Right panel: *TD2/ PsiRNAi* (*Psi* knockdown) flies. Note the short period of *Psi* knockdown flies. n = 8 flies/genotype. (**C–E**) Circadian period length (hrs) is plotted on the y

*Figure 2 continued on next page*

Figure 2 continued

axis. Genotypes are listed on the x axis. Error bars represent SEM. Solid black bar is $w^{1118}$ (WT) control; solid blue, magenta and gray bars are driver controls; patterned bars are *Psi* knockdown with two non-overlapping RNAi lines: *GD14067* (*PsiRNAiGD*) and *KK101882* (*PsiRNAiKK*). *p<0.05, ***p<0.001, ****p<0.0001, one-way ANOVA followed by Tukey's multiple comparison test (C) Dunnett's multiple comparison test (D and E). (C) Knockdown in all circadian tissues. Left panel 25˚C, right panel 30˚C. Note that even at 25˚C, the experimental flies are shorter than their respective *RNAi/+* control, despite the dominant period lengthening caused by *TD2* (D) Knockdown in PDF+ circadian pacemaker neurons. (E) Knockdown in PDF- circadian tissues. In D and E, only the driver controls are shown, since they are the controls which the experimental flies need to be compared to because of the dominant period lengthening caused by *PD2* and *TD2*. (F–H) Overexpression of *Psi* lengthens the behavioral period and decreases rhythmicity. Left panels: Circadian period length (hrs) is plotted on the y axis. Error bars represent SEM. Right panels: Percent of flies that remained rhythmic in DD is plotted on the y axis. Both panels: Genotypes are listed on the x axis. Not significant (ns)p>0.05, *p<0.05, ****p<0.0001, one-way ANOVA followed by Tukey's multiple comparison test. (F) Overexpression of *Psi* in all circadian tissues lengthened the circadian period and decreased the percent of rhythmic flies. (G) Overexpression of *Psi* in PDF+ circadian pacemaker neurons caused a slight but non-significant period lengthening compared to the driver control (*PG4/+*), which is the relevant comparison because of the dominant period lengthening caused by *PG4*. Rhythmicity was slightly reduced compared to *PG4/+* but not compared to *UAS-Psi/+*. (H) Overexpression of *Psi* in PDF- circadian tissues lengthened the circadian period and decreased rhythmicity.

The online version of this article includes the following source data and figure supplement(s) for figure 2:

**Source data 1.** *Psi* downregulation and overexpression – behavior data.
**Source data 2.** Figure statistics – *Figure 2*.
**Figure supplement 1.** *Psi* mRNA expression does not cycle and its level is reduced in heads of *Psi* knockdown flies.
**Figure supplement 1—source data 1.** *Psi* qPCR data.
**Figure supplement 1—source data 2.** Figure statistics – *Figure 2—figure supplement 1*.
**Figure supplement 2.** Knockdown of Psi shortens circadian period of PER and TIM rhythms in peripheral tissues.
**Figure supplement 2—source data 1.** TIMLUC signal.
**Figure supplement 2—source data 2.** BGLUC signal.
**Figure supplement 2—source data 3.** Figure statistics – *Figure 2—figure supplement 2*.

knockdown fly heads was significantly reduced at both CT 9 and CT 15, indicating that the RC isoform's contribution to total *cwo* mRNA levels is quite modest (*Figure 3—figure supplement 1E*).

We then analyzed alternative splicing of *tim* in *Psi* knockdown heads compared to controls. Specifically, we looked at the expression of three temperature-sensitive intron inclusion events in *tim* that all theoretically lead to C-terminal truncations of the protein (*Figure 3A*). The *tim-cold* isoform, which is not annotated in Flybase (*Thurmond et al., 2019*), is dominant at low temperature (18˚C) and arises when the last intron is retained (*Boothroyd et al., 2007*). We found that *tim-cold* is elevated in *Psi* knockdown heads at peak levels under 25˚C conditions (CT15, *Figure 3D*). Similarly, we found that another intron inclusion event, *tim-sc (tim-short and cold)* which has also been shown to be elevated at 18˚C and is present in the *tim-RN* and *-RO* isoforms (*Martin Anduaga et al., 2019*), is significantly increased at 25˚C in *Psi* knockdown heads at CT15 (*Figure 3B*). Thus, interestingly, two intron inclusion events that are upregulated by cold temperature are also both upregulated in *Psi* knockdown heads at 25˚C. In contrast, we found that an intron included in the *tim-RM* and *-RS* isoforms (*tim-M, for tim-Medium)* and shown to be increased at high temperature (29˚C, *Martin Anduaga et al., 2019*; *Shakhmantsir et al., 2018*) is significantly decreased at CT 9, 15 and 21 in *Psi* knockdown heads at 25˚C (*Figure 3F*). In the case of *tim-sc,* it should be noted that the intron is only partially retained, because a cleavage and poly-adenylation signal is located within this intron, thus resulting in a much shorter mature transcript (*Martin Anduaga et al., 2019*). Based on PSI function, the most parsimonious explanation is that PSI reduces production of *tim-sc* by promoting splicing of the relevant intron. However, we cannot entirely exclude that PSI regulates the probability of premature cleavage causing the RNA polymerase to undergo transcription termination soon after passing the poly-adenylation signal.

Collectively, these results indicate that, in wild-type flies, PSI shifts the balance of *tim* alternative splicing events toward a warm temperature *tim* RNA isoform profile at an intermediate temperature (25˚C). This could be achieved either by altering the temperature sensitivity of *tim* introns, or by promoting a 'warm temperature splicing pattern' independently of temperature. We therefore also measured *tim* splicing isoforms at 18˚C and 29˚C (*Figure 3C,E,G*). We entrained flies for 3 days in LD at 25˚C to maintain similar levels of GAL4 expression and thus of *Psi* knockdown (the GAL4/UAS system's activity increases with temperature, *Duffy, 2002*). We then shifted them to either 18˚C or 29˚C

**Table 3.** PSI affects circadian behavior

| Genotype | Period ±SEM | Power ±SEM | n | % of Rhythmic Flies |
|---|---|---|---|---|
| *Psi* downregulation and overexpression at 25°C | | | | |
| TD2/+ | 24.8 ± 0.04 | 48.2 ± 2.3 | 71 | 82 |
| TD2/PsiRNAiGD | 23.7 ± 0.07 | 49.6 ± 3.0 | 48 | 79 |
| TD2/PsiRNAiKK | 23.6 ± 0.06 | 61.9 ± 3.7 | 35 | 77 |
| PD2/+ | 24.9 ± 0.04 | 50.4 ± 2.1 | 77 | 83 |
| PD2/PsiRNAiGD | 24.2 ± 0.06 | 53.3 ± 4.1 | 32 | 84 |
| PD2/PsiRNAiKK | 24.7 ± 0.06 | 56.3 ± 3.4 | 47 | 89 |
| TD2/+; PdfGAL80/+ | 24.5 ± 0.07 | 49.4 ± 2.8 | 40 | 75 |
| TD2/PsiRNAiGD; PdfGAL80/+ | 23.8 ± 0.17 | 45.8 ± 5.5 | 24 | 50 |
| TD2/PsiRNAiKK; PdfGAL80/+ | 24.0 ± 0.05 | 71.9 ± 4.0 | 39 | 95 |
| $w^{1118}$ | 24.1 ± 0.03 | 84.8 ± 2.5 | 70 | 99 |
| PsiRNAiGD/+ | 24.2 ± 0.04 | 58.9 ± 2.9 | 63 | 94 |
| PsiRNAiKK/+ | 24.0 ± 0.04 | 67.1 ± 3.7 | 55 | 96 |
| TG4/+ | 25.2 ± 0.05 | 52.5 ± 2.2 | 68 | 88 |
| TG4/+; UAS-Psi/+ | 25.9 ± 0.07 | 31.3 ± 1.2 | 302 | 16 |
| PG4/+ | 25.0 ± 0.05 | 66.0 ± 3.5 | 26 | 96 |
| PG4/+; UAS-Psi/+ | 25.2 ± 0.07 | 44.0 ± 2.7 | 48 | 77 |
| TG4/+; PdfGAL80/+ | 24.6 ± 0.06 | 42.8 ± 2.8 | 37 | 84 |
| TG4/+; PdfGAL80/UAS-Psi | 24.9 ± 0.19 | 31.3 ± 2.8 | 116 | 11 |
| UAS-Psi/+ | 24.2 ± 0.04 | 46.4 ± 1.8 | 80 | 79 |
| *Psi* downregulation at 20°C | | | | |
| TD2/+ | 24.9 ± 0.10 | 42.0 ± 3.1 | 39 | 59 |
| TD2/PsiRNAiGD | 23.6 ± 0.07 | 52.2 ± 4.7 | 44 | 66 |
| TD2/PsiRNAiKK | 23.7 ± 0.08 | 43.8 ± 5.5 | 44 | 36 |
| PsiRNAiGD/+ | 24.0 ± 0.09 | 46.0 ± 3.7 | 32 | 72 |
| PsiRNAiKK/+ | 23.8 ± 0.08 | 39.1 ± 4.9 | 32 | 38 |
| *Psi* downregulation at 30°C | | | | |
| TD2/+ | 23.7 ± 0.07 | 48.2 ± 2.9 | 39 | 87 |
| TD2/PsiRNAiGD | 23.1 ± 0.13 | 38.3 ± 3.8 | 42 | 40 |
| TD2/PsiRNAiKK | 22.8 ± 0.15 | 43.1 ± 4.2 | 41 | 41 |
| PsiRNAiGD/+ | 23.6 ± 0.04 | 43.2 ± 3.4 | 32 | 75 |
| PsiRNAiKK/+ | 23.5 ± 0.03 | 63.0 ± 3.7 | 31 | 90 |
| TIM-HA suppression of PSI's effect on circadian behavior | | | | |
| TG4/PsiRNAiKK; UAS-Dcr2/+ | 23.4 ± 0.04 | 59.5 ± 4.3 | 57 | 75 |
| TG4/+; UAS-Dcr2/+ | 24.9 ± 0.04 | 59.4 ± 3.1 | 36 | 92 |
| $tim^0$,TG4/$tim^0$; UAS-Dcr2/timHA | 24.9 ± 0.07 | 44.3 ± 4.0 | 28 | 75 |
| $tim^0$,TG4/$tim^0$,PsiRNAiKK; UAS-Dcr2/timHA | 24.8 ± 0.06 | 50.0 ± 2.9 | 38 | 79 |

at CT 0 on the first day of DD and collected samples at CT 3, 9, 15 and 21. We found that both the *tim-cold* intron and the *tim-sc* introns were elevated at 18°C in both *Psi* knockdown heads and controls (*Figure 3C and E*). Thus, *Psi* knockdown does not block the temperature sensitivity of these introns. *tim-M* levels were unexpectedly variable in DD, particularly in the *Psi* knockdown flies, perhaps because of the temperature change. Nevertheless, we observed a trend for the *tim-M* intron retention to be elevated at 29°C (*Figure 3G*), further supporting our conclusion that *Psi* knockdown

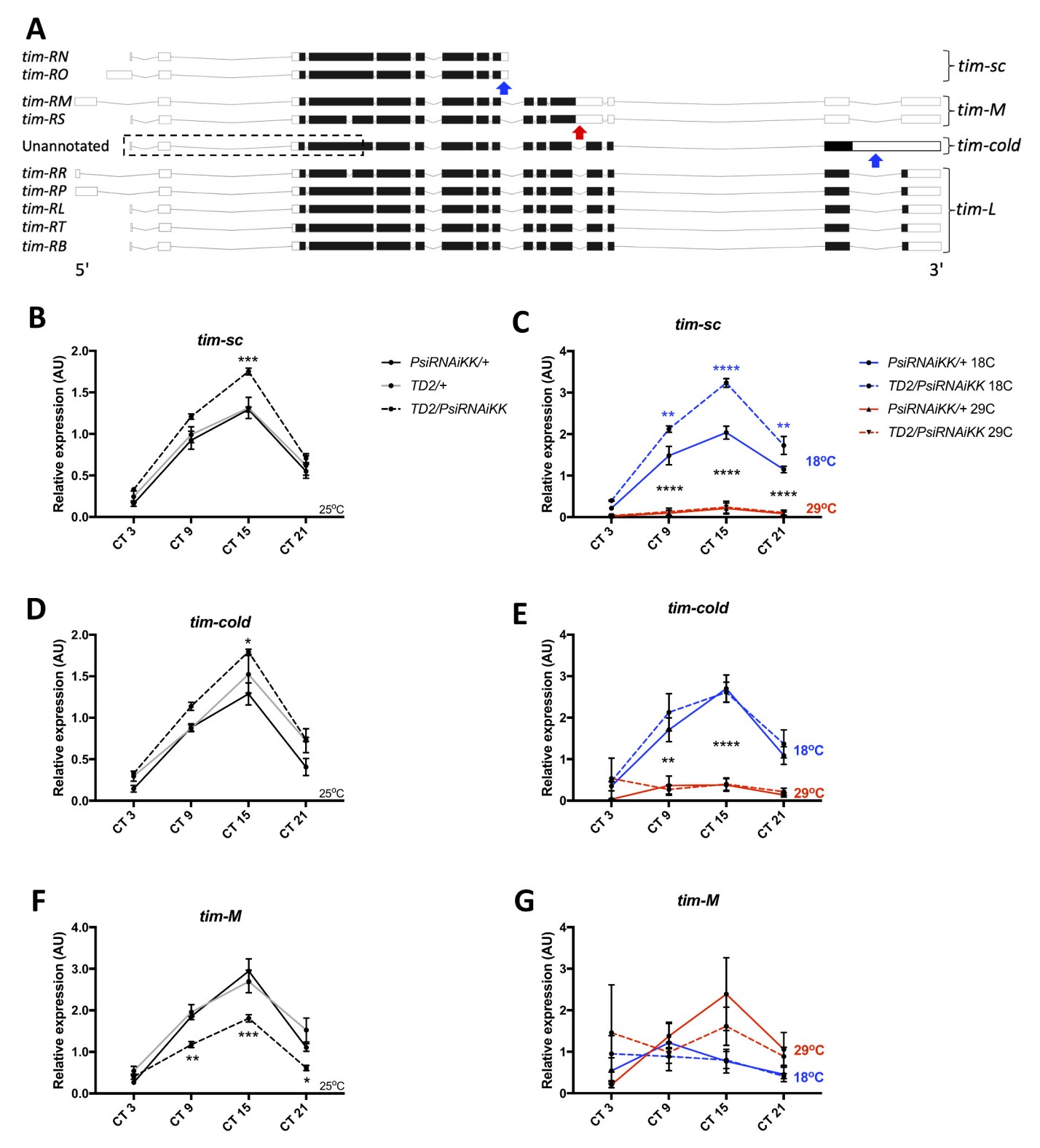

**Figure 3.** Knockdown of Psi increases the expression of cold induced *tim* isoforms and decreases the expression of a warm induced *tim* isoform. (**A**) Schematic of *tim* isoforms. Flybase transcript nomenclature on left, intron retention events studied here on right (*tim-L* refers to *tim* transcripts that do not produce C-terminal truncations of TIM via intron retention). Arrows indicate the location of retained introns: blue, upregulated at cold temperature; red, upregulated at warm temperature. The retained intron that gives rise to the *tim-cold* isoform is not annotated in Flybase (***Thurmond et al., 2019***). It is possible that multiple *tim-cold* transcripts may exist due to alternative splicing and alternative transcription/translation start sites in the 5' region of

*Figure 3 continued on next page*

*Figure 3 continued*

the gene (dashed box). However, for simplicity, we depict this region of *tim-cold* using the most common exons. Adapted from Ensembl 94 (*Zerbino et al., 2018*). (B, D, F) Relative expression of *tim* mRNA isoforms at 25°C (normalized to the average of all *Psi* knockdown time points) in heads on the y axis measured by qPCR. Circadian time (CT) on the x axis. Error bars represent SEM. Gray line: driver control. Black line: RNAi control. Dashed line: *Psi* knockdown. Controls, N = 3. *Psi* knockdown, N = 5 (3 technical replicates per sample). Both driver and RNAi control compared to *Psi* knockdown, two-way ANOVA followed by Tukey's multiple comparison test: *p<0.05, **p<0.01, ***p<0.001, ****p<0.0001. (C, E, G) Relative expression of *tim* mRNA isoforms at 18°C and 29°C (normalized to the average of all *Psi* knockdown time points). Solid line: RNAi control. Dashed line: *Psi* RNAi knockdown. Blue indicates flies were transferred to 18°C at CT0 (start of subjective day) on the first day of DD. Red indicates flies were transferred to 29°C. N = 3 (3 technical replicates per sample). 18°C samples compared to 29°C samples, *p<0.05, **p<0.01, ***p<0.001, ****p<0.0001, two-way ANOVA followed by Tukey's multiple comparison test. (C) Blue asterisks refer to RNAi control compared to *Psi* knockdown.

The online version of this article includes the following source data and figure supplement(s) for figure 3:

**Source data 1.** *tim* qPCR data.
**Source data 2.** Figure statistics – *Figure 3*.
**Figure supplement 1.** Knockdown of Psi affects the balance of *cwo* isoform expression.
**Figure supplement 1—source data 1.** *cwo* qPCR data.
**Figure supplement 1—source data 2.** Figure statistics – *Figure 3—figure supplement 1*.
**Figure supplement 2.** *Psi* knockdown flies have normal behavioral adaptation to temperature.
**Figure supplement 2—source data 1.** *Psi* downregulation – anticipation phase.
**Figure supplement 2—source data 2.** Figure statistics – *Figure 3—figure supplement 2*.
**Figure supplement 3.** *Psi* knockdown flies have a normal photic phase response.
**Figure supplement 3—source data 1.** *Psi* downregulation – PRC.
**Figure supplement 3—source data 2.** Figure statistics – *Figure 3—figure supplement 3*.

does not affect the temperature sensitivity of *tim* splicing, but rather determines the ratio of *tim* mRNA isoforms, and it does this at all temperatures.

As expected from these results, *Psi* downregulation did not affect the ability of flies to adjust the phase of their evening and morning peak to changes in temperature (*Figure 3—figure supplement 2*). We also tested whether *Psi* knockdown flies responded normally to short light pulses, since TIM is the target of the circadian photoreceptor CRY (*Emery et al., 1998*; *Stanewsky et al., 1998*; *Lin et al., 2001*; *Busza et al., 2004*; *Koh et al., 2006*). These flies could both delay or advance the phase of their circadian behavior in response to early or late-night light pulses, respectively (*Figure 3—figure supplement 3*). We noticed however a possible slight shift of the whole Phase Response Curve toward earlier times. This would be expected since the pace of the circadian clock is accelerated.

## PSI controls the phase of circadian behavior under temperature cycle

Since PSI regulates thermosensitive *tim* splicing events, we wondered whether it might have an impact on circadian behavioral responses to temperature. As mentioned above, *Psi* downregulation does not affect *Drosophila*'s ability to adjust the phase of their behavior to different constant ambient temperatures, under a LD cycle (*Figure 3—figure supplement 2*). *Psi* knockdown did not appear to affect temperature compensation, as these flies essentially responded to temperature in a similar way as their *TD2/+* control, with shorter period at 29°C (*Figure 4—figure supplement 1*). However, we found a striking phenotype in flies with *Psi* downregulation under temperature cycle (29/20°C). Once flies had reached a stable phase relationship with the entraining temperature cycle (*Busza et al., 2007*), the phase of the evening peak of activity was advanced by about 2.5 hr in *TD2/ PsiRNAi*, compared to controls, and this with two non-overlapping dsRNAs (*Figure 4*). Controls included *TD2/+* or *TD2/VIE-260B* (KK host strain), *RNAi/+*, as well as *TD2* crossed to a KK or GD RNAi line that did not produce a circadian phenotype. Importantly, no such phase advance was observed under LD (*Figure 3—figure supplement 2*), indicating that the short period phenotype does not account for the evening-peak advanced phase under temperature cycle. Rather, the phase advance is specific to temperature entrainment. The morning peak was difficult to quantify as it tended to be of low amplitude.

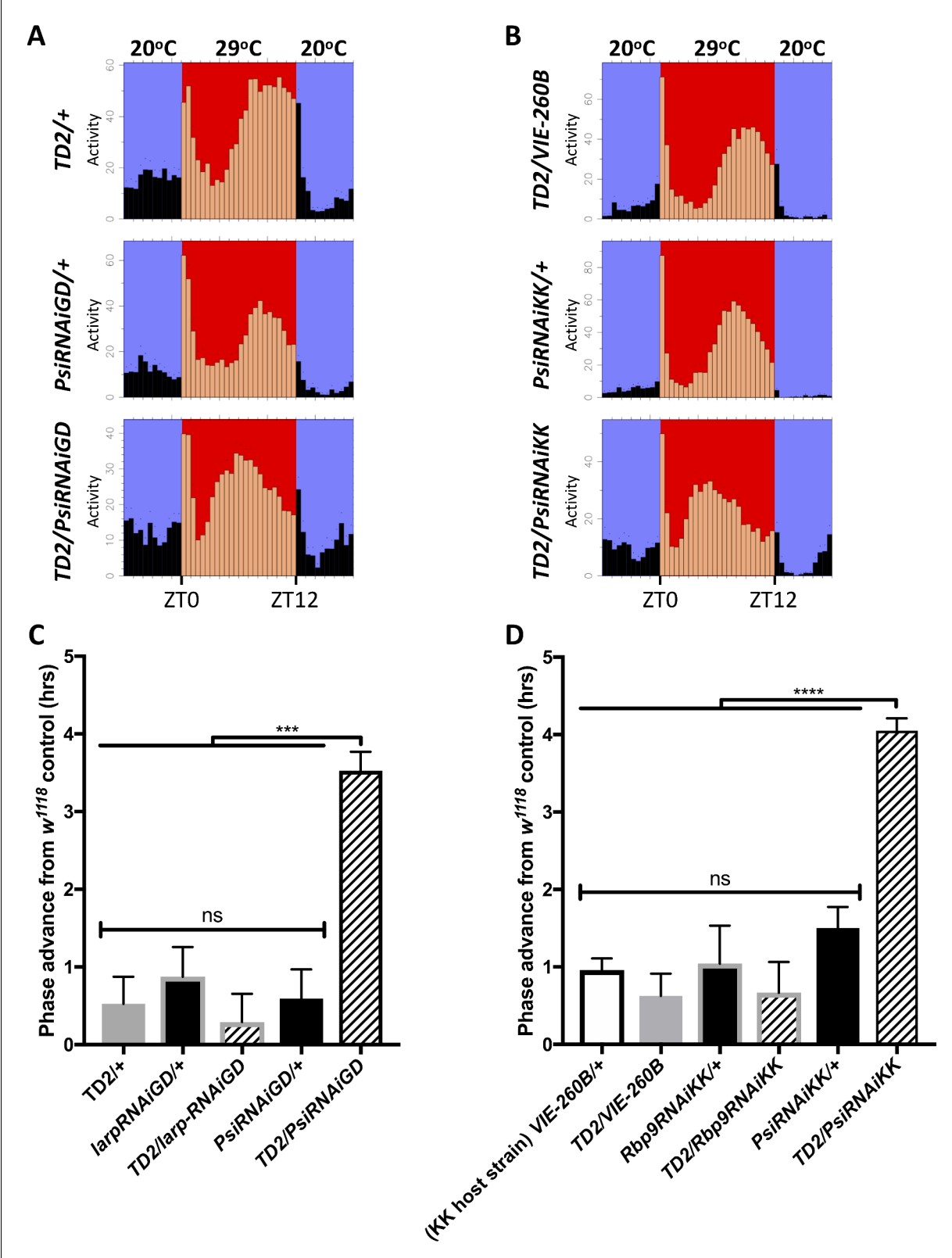

**Figure 4.** Knockdown of Psi advances the phase of circadian behavior under temperature cycle. (**A**) Eductions showing the average activity of flies during 4 days of 12:12 29°C(red)/20°C(blue) temperature entrainment (days 7–10) in DD. Top panels: (driver controls) *TD2/+* (left), *TD2/VIE-260B* (right). Middle panels: (RNAi controls) *PsiRNAiGD/+* (left), *PsiRNAiKK/+* (right). Bottom panels: (*Psi* knockdown) *TD2/PsiRNAiGD* (left), *TD2/PsiRNAiKK* (right). Note that, *Psi* knockdown flies advance the phase of their evening activity by about 2.5 hr relative to controls. (**C–D**) Evening peak phase relative to an

*Figure 4 continued on next page*

Figure 4 continued

internal control in each run ($w^{1118}$) (hrs) is plotted on the y axis. Genotypes are listed on the x axis. Error bars represent SEM. ***p<0.001, ****p<0.0001, one-way ANOVA followed by Tukey's multiple comparison test. N = 3–5 runs (C) Quantification of PsiRNAiGD knockdown and controls. Note additional RNAi controls: larpRNAiGD/+ (black bar, gray border) and TD2/larpRNAiGD (patterned bar, gray border). larpRNAiGD (GD8214) is an RNAi line from the GD collection that targets a RAP from our screen that was not a hit. (D) Quantification of PsiRNAiKK knockdown and controls. Note additional RNAi controls: VIE260B/+ (white bar, black border), TD2/VIE260B (gray bar), Rbp9RNAiKK/+ (black bar, gray border) and TD2/Rbp9RNAiKK (patterned bar, gray border). VIE260B is a KK collection host strain control containing the 30B transgene insertion site. Rbp9RNAiKK (KK109093) is an RNAi line from the KK collection targeting a RAP from our screen that was not a hit.

The online version of this article includes the following source data and figure supplement(s) for figure 4:

**Source data 1.** *Psi* downregulation – temperature cycle phase.
**Source data 2.** Figure statistics – *Figure 4*.
**Figure supplement 1.** Free-running circadian behavior of *Psi* knockdown flies and controls at different temperatures in DD.
**Figure supplement 1—source data 1.** *Psi* downregulation – temperature compensation.
**Figure supplement 1—source data 2.** Figure statistics – *Figure 4—figure supplement 1*.

## *tim* splicing is required for PSI's regulation of circadian period and circadian behavior phase under temperature cycle

Because *tim* is a key element of the circadian transcriptional feedback loop and its splicing pattern is determined by the ambient temperature, we wondered whether PSI might be regulating the speed of the clock and the phase of the evening peak through its effects on *tim* splicing. We therefore rescued the amorphic *tim* allele ($tim^0$) with a *tim* transgene that lacks the known temperature sensitive alternatively spliced introns as well as most other introns (timHA) (*Figure 5A*) (*Rutila et al., 1998*). Importantly, the $tim^0$ mutation is a frame-shifting deletion located upstream of the temperature-sensitive alternative splicing events (*Myers et al., 1995*), and would thus truncate any TIM protein produced from the splice variants we studied. Strikingly, we found that knockdown of *Psi* in timHA rescued $tim^0$ flies had no impact on the period of circadian behavior (*Figure 5B–C*, *Table 3*). Likewise, the evening peak phase under temperature cycles was essentially insensitive to *Psi* knockdown in timHA rescued $tim^0$ flies (*Figure 5D–E*). This indicates that PSI controls circadian period in DD and the phase of the evening peak under temperature cycle through *tim* splicing.

## Discussion

Our results identify a novel post-transcriptional regulator of the circadian clock: PSI. PSI is required for the proper pace of both brain and body clock, and for proper phase-relationship with ambient temperature cycles. When *Psi* is downregulated, the circadian pacemaker speeds up and behavior phase under temperature cycles is advanced by 3 hr, and these phenotypes appear to be predominantly caused by an abnormal *tim* splicing pattern. Indeed, the circadian period and behavior phase of flies that can only produce functional TIM protein from a transgene missing most introns is insensitive to *Psi* downregulation. We note however that *cwo*'s splicing pattern is also affected by *Psi* downregulation, and we did not study *sgg* splicing pattern, although it might also be controlled by PSI (*Wang et al., 2016*). We therefore cannot exclude a small contribution of non-*tim* splicing events to PSI downregulation phenotypes, or that in specific tissues these other splicing events play a greater role than in the brain.

Interestingly, *Psi* downregulation results in an increase in intron inclusion events that are favored under cold conditions (*tim-sc* and *tim-cold*), while an intron inclusion event favored under warm conditions is decreased (*tim-M*). However, the ability of *tim* splicing to respond to temperature changes is not abolished when *Psi* is downregulated (*Figure 3C,E,G*). This could imply that an as yet unknown factor specifically promotes or represses *tim* splicing events in a temperature-dependent manner. Another possibility is that the strength of splice sites or *tim*'s pre-mRNA structure impacts splicing efficiency in a temperature–dependent manner. For example, suboptimal *per* splicing signals explain the lower efficiency of *per*'s most 3' splicing event at warm temperature (*Low et al., 2008*).

How would the patterns of *tim* splicing affect the pace of the circadian clock, or advance the phase of circadian behavior under temperature cycles? In all splicing events that we studied, intron retention results in a truncated TIM protein. It is therefore possible that the balance of full length and truncated TIM proteins, which may function as endogenous dominant-negatives, determines

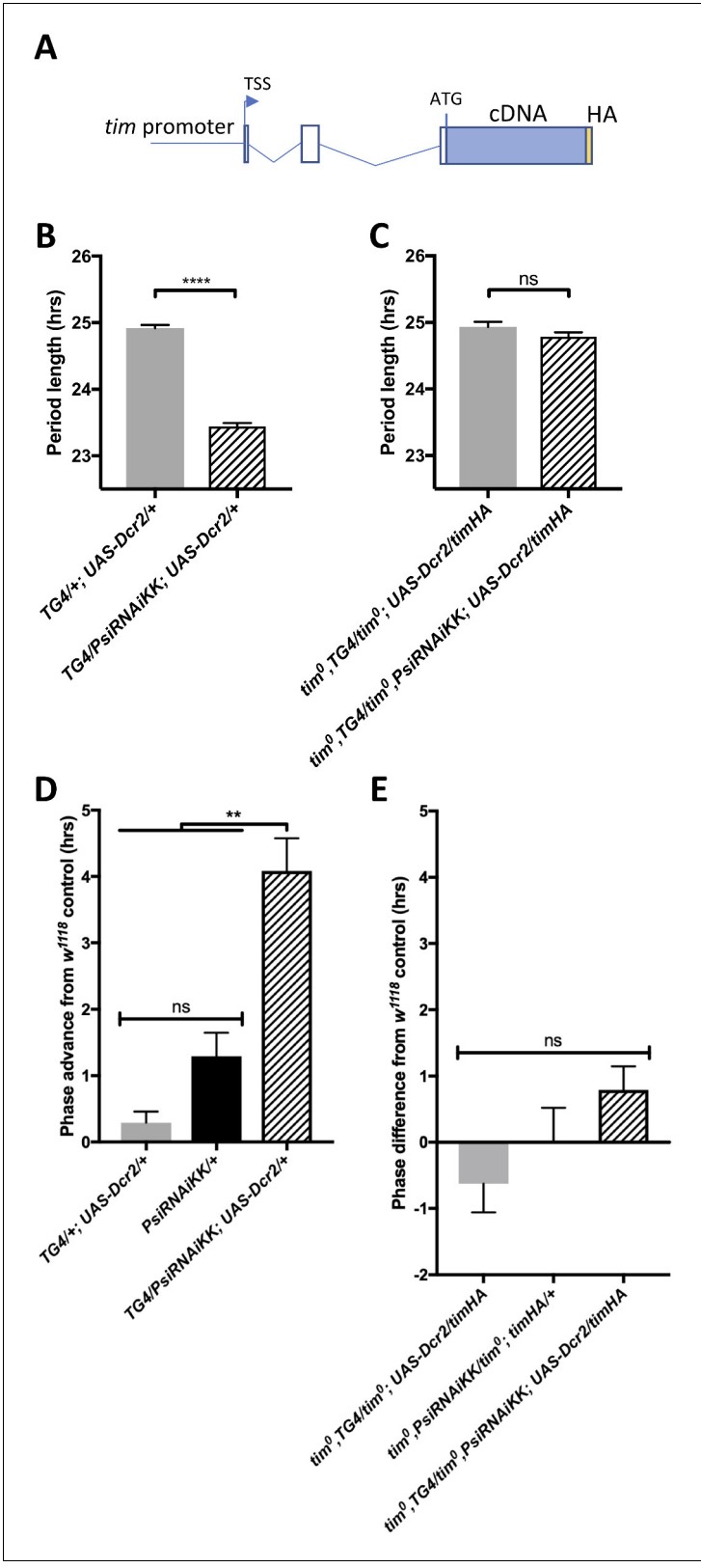

**Figure 5.** The short period and temperature cycle phase advance effects of Psi knockdown are dependent on *tim* introns. (**A**) Schematic of *timHA* transgene. The *tim* promoter is fused upstream of the transcription start site (TSS). Two introns remain in the 5'UTR, upstream of the start codon; however, they are not, to our knowledge, temperature sensitive. A C-terminal HA tag is fused to full length *tim* cDNA, which lacks any of the introns that are
*Figure 5 continued on next page*

*Figure 5 continued*

known to be retained at high or low temperatures. (**B**) Knockdown of *Psi* with *tim-GAL4* and a *UAS-dcr2* transgene inserted on the 3$^{rd}$ chromosome also causes period shortening. We used this insertion to more easily generate stocks in a *tim$^0$* background, since the *tim* gene is on the second chromosome, instead of the TD2 combination that has both the *tim-GAL4 and UAS-dcr2* transgenes on the 2$^{nd}$ chromosome. ****p<0.0001, Student's t-test. (**C**) Period shortening in response to *Psi* knockdown with *tim-GAL4 and UAS-dcr* is abolished in *tim$^0$, ptim-timHA* flies that can only produce the full length *tim* isoform. ns, p=0.1531, Student's t-test. (**B, C**) Circadian period length (hrs) is plotted on the y axis. Genotypes are listed on the x axis. Error bars represent SEM. (**D**) Knockdown of *Psi* with *tim-GAL4* and a *UAS-dcr2* 3$^{rd}$ chromosome transgene also causes a phase advance in a 12:12 29°C/20°C temperature cycle. (**E**) The phase advance is abolished in *tim$^0$, ptim-timHA* flies that can only produce the full length *tim* isoform. (**D, E**) Evening peak phase relative to an internal control in each run (*w$^{1118}$*) (hrs) is plotted on the y axis. Genotypes are listed on the x axis. Error bars represent SEM. **p<0.01, one-way ANOVA followed by Tukey's multiple comparison test. N = 3 runs.

The online version of this article includes the following source data for figure 5:

**Source data 1.** *Psi* downregulation in a *tim$^0$; timHA* background – behavioral period length in DD and temperature cycle phase.

**Source data 2.** Figure statistics – *Figure 5*.

---

circadian period. For example, truncated TIM might be less efficient at protecting PER from degradation, thus accelerating the pacemaker, or affecting its phase. Consistent with this idea, overexpression of the shorter cold-favored *tim* isoform (*tim-sc*) shortens period (*Martin Anduaga et al., 2019*). Strikingly, *Psi* downregulation increases this isoform's levels and also results in a short phenotype. *Shakhmantsir et al. (2018)* also proposed that production of *tim-M* transcripts (called *tim-tiny* in their study) delays the rate of TIM accumulation. Such a mechanism could also contribute to the short period we observed when *Psi* is downregulated, since this reduces *tim-M* levels, which may accelerate TIM accumulation. Another interesting question is how PSI differentially affects specific splice isoforms of *tim*. One possibility is that the execution of a specific *tim* splicing event negatively influences the probability of the occurrence of other splicing events. For example, PSI could downregulate *tim-sc* and *tim-cold* by enhancing splicing and removal of the introns whose retention is necessary for production of these isoforms. This could indirectly reduce splicing of the intron that is retained in the warm *tim-M* isoform and result in *tim-M* upregulation. Conversely, PSI could directly promote *tim-M* intron retention and indirectly downregulate production of *tim-sc* and *tim-cold*.

Other splicing factors have been shown to be involved in the control of circadian rhythms in *Drosophila*. SRm160 contributes to the amplitude of circadian rhythms by promoting *per* expression (*Beckwith et al., 2017*), while B52/SMp55 and PRMT5 regulate *per*'s most 3' splicing, which is temperature sensitive (*Zhang et al., 2018*; *Sanchez et al., 2010*). Loss of PRMT5 results in essentially arrhythmic behavior (*Sanchez et al., 2010*), but this is unlikely to be explained by its effect on *per*'s thermosensitive splicing. B52/SMp55 knockdown flies show a reduced siesta, which is controlled by the same *per* splicing (*Zhang et al., 2018*). With the identification of *Psi*, we uncover a key regulator of *tim* alternative splicing pattern and show that this pattern determines circadian period length, while *per* alternative splicing regulates the timing and amplitude of the daytime siesta. Interestingly, a recent study identified PRP4 kinase and other members of tri-snRNP complexes as regulators of circadian rhythms (*Shakhmantsir et al., 2018*). Downregulation of *prp4* caused excessive retention of the *tim-M* intron. PSI and PRP4 might thus have complementary functions in *tim* mRNA splicing regulation, working together to maintain the proper balance of *tim* isoform expression.

An unexpected finding is the role played by both PDF neurons and other circadian neurons in the short period phenotype observed with circadian locomotor rhythms when we knocked-down *Psi*. Indeed, it is quite clear from multiple studies that under constant darkness, the PDF-positive sLNvs dictate the pace of circadian behavior (*Stoleru et al., 2005*; *Yao and Shafer, 2014*). Why, in the case of *Psi* downregulation, do PDF negative neurons also play a role in period determination? The explanation might be that PSI alters the hierarchy between circadian neurons, promoting the role of PDF negative neurons. This could be achieved by weakening PDF/PDFR signaling, for example.

While we focused our work on PSI, several other interesting candidates were identified in our screen (*Tables 1* and *2*). We note the presence of a large number of splicing factors. This adds to the emerging notion that alternative splicing plays a critical role in the control of circadian rhythms.

We have already mentioned above several *per* splicing regulators that can impact circadian behavior. In addition, a recent study demonstrated that specific classes of circadian neurons express specific alternative splicing variants, and that rhythmic alternative splicing is widespread in these neurons (*Wang et al., 2018*). Interestingly, in this study, the splicing regulator *barc*, which was identified in our screen and which has been shown to causes intron retention in specific mRNAs (*Abramczuk et al., 2017*), was found to be rhythmically expressed in LNds. Moreover, in mammals, alternative splicing appears to be very sensitive to temperature, and could explain how body temperature rhythms synchronize peripheral clocks (*Preußner et al., 2017*). Another intriguing candidate is *cg42458*, which was found to be enriched in circadian neurons (LNvs and Dorsal Neurons 1) (*Wang et al., 2018*). In addition to emphasizing the role of splicing, our screen suggests that regulation of polyA tail length is important for circadian rhythmicity, since we identified several members of the CCR4-NOT complex and deadenylation-dependent decapping enzymes. Future work will be required to determine whether these factors directly target mRNAs encoding for core clock components, or whether their effect on circadian period is indirect. Interestingly, the POP2 deadenylase, which is part of the CCR4-NOT complex, was recently shown to regulate *tim* mRNA levels post-transcriptionally (*Grima et al., 2019*). It should be noted that while our screen targeted 364 proteins binding or associated with RNA, it did not include all of them. For example, LSM12, which was recently shown to be a part of the ATXN2/TYF complex (*Lee et al., 2017*), was not included in our screen because it had not been annotated as a potential RAP when we initiated our screen.

In summary, our work provides an important resource for identifying RNA associated proteins regulating circadian rhythms in *Drosophila*. It identifies PSI is an important regulator of circadian period and circadian phase in response to thermal cycles, and points at additional candidates and processes that determine the periodicity of circadian rhythms.

## Materials and methods

**Key resources table**

| Reagent type (species) or resource | Designation | Source or reference | Identifiers | Additional information |
|---|---|---|---|---|
| Gene (*Drosophila melanogaster*) | *Psi* | | FLYB:FBgn0014870 | Flybase name: *P-element somatic inhibitor* |
| Gene (*Drosophila melanogaster*) | *tim* | | FLYB:FBgn0014396 | Flybase name: *timeless* |
| Gene (*Drosophila melanogaster*) | *tio* | | FLYB:FBgn0028979 | Flybase name: *tiptop* |
| Gene (*Drosophila melanogaster*) | *per* | | FLYB:FBgn0003068 | Flybase name: *period* |
| Gene (*Drosophila melanogaster*) | *cwo* | | FLYB:FBgn0259938 | Flybase name: *clockwork orange* |
| Gene (*Drosophila melanogaster*) | *RpL32* | | FLYB:FBgn0002626 | qPCR control Flybase name: *Ribosomal protein L32* |
| Gene (*Drosophila melanogaster*) | *larp* | | FLYB:FBgn0261618 | Flybase name: *La related protein* |
| Gene (*Drosophila melanogaster*) | *Rbp9* | | FLYB:FBgn0010263 | Flybase name: *RNA-binding protein 9* |
| Gene (*Drosophila melanogaster*) | *Dcr-2* | | FBgn0034246 | Flybase name: *Dicer-2* |

*Continued on next page*

*Continued*

| Reagent type (species) or resource | Designation | Source or reference | Identifiers | Additional information |
|---|---|---|---|---|
| Genetic reagent (*D. melanogaster*) | *tim-GAL4* | *Kaneko et al., 2000* | FLYB:FBtp0010385 | |
| Genetic reagent (*D. melanogaster*) | *Pdf-GAL4* | *Renn et al., 1999* | FLYB:FBtp0011844 | |
| Genetic reagent (*D. melanogaster*) | *Pdf-GAL80, Pdf-GAL80* | *Stoleru et al., 2004* | | |
| Genetic reagent (*D. melanogaster*) | *UAS-Dcr2* | *Dietzl et al., 2007* | FLYB:FBti0100275 RRID:BDSC_24650 | Chromosome 2 |
| Genetic reagent (*D. melanogaster*) | *UAS-Dcr2* | *Dietzl et al., 2007* | FLYB:FBti0100276 | Chromosome 3 |
| Genetic reagent (*D. melanogaster*) | *PsiRNAi KK101882* | | FLYB:FBal0231542 | |
| Genetic reagent (*D. melanogaster*) | *PsiRNAi GD14067* | *Dietzl et al., 2007* | FLYB:FBst0457756 | |
| Genetic reagent (*D. melanogaster*) | *UAS-Psi* | *Labourier et al., 2001* | | |
| Genetic reagent (*D. melanogaster*) | *BG-LUC* | *Stanewsky et al., 1997* | | |
| Genetic reagent (*D. melanogaster*) | *ptim-TIMLUC* | *Lamba et al., 2018* | | |
| Genetic reagent (*D. melanogaster*) | *timHA* | *Rutila et al., 1998* | FLYB:FBal0143160 | |
| Genetic reagent (*D. melanogaster*) | *tim^0* | *Sehgal et al., 1994* | FLYB:FBal0035778 | |
| Genetic reagent (*D. melanogaster*) | *VIE260B* | | VDRC_ID: 60100 | |
| genetic reagent (*D. melanogaster*) | *larpRNAi GD8214* | *Dietzl et al., 2007* | VDRC_ID: 17366 | |
| Genetic reagent (*D. melanogaster*) | *Rbp9RNAi KK109093* | | VDRC_ID: 101412 | |
| Genetic reagent (*D. melanogaster*) | *w^1118* | | VDRC_ID: 60000 | |
| Genetic reagent (*D. melanogaster*) | *40D-UAS* | | VDRC_ID: 60101 | |
| Sequence-based reagent | *RpL32*-forward | *Dubruille et al., 2009* | PCR primers | ATGCTAAGCTGTCGCACAAA |
| Sequence-based reagent | *RpL32*-reverse | *Dubruille et al., 2009* | PCR primers | GTTCGATCCGTAACCGATGT |
| Sequence-based reagent | *psi*-forward | This paper | PCR primers | GGTGCCTTGAATGGGTGAT |

*Continued on next page*

*Continued*

| Reagent type (species) or resource | Designation | Source or reference | Identifiers | Additional information |
|---|---|---|---|---|
| Sequence-based reagent | *psi*-reverse | This paper | PCR primers | CGATTTATCCGGGTCCTCG |
| Sequence-based reagent | *tim-M*-forward | This paper | PCR primers | TGGGAATCTCGCCCGAAAC |
| Sequence-based reagent | *tim-M*-reverse | This paper | PCR primers | AGAAGGAGGAGAAGGAGAGAGG |
| Sequence-based reagent | *tim-sc*-forward | This paper | PCR primers | ACTGTGCGATGACTGGTCTG |
| Sequence-based reagent | *tim-sc*-reverse | This paper | PCR primers | TGCTTCAAGGAAATCTTCTG |
| Sequence-based reagent | *tim-cold*-forward | This paper | PCR primers | CCTCCATGAAGTCCTCGTTCG |
| Sequence-based reagent | *tim-cold*-reverse | This paper | PCR primers | ATTGAGCTGGGACACCAGG |
| Sequence-based reagent | *cwo*-foward | This paper | PCR primers | TTCCGCTGTCCACCAACTC |
| Sequence-based reagent | *cwo*-reverse | This paper | PCR primers | CGATTGCTTTGCTTTACCAGCTC |
| Sequence-based reagent | *cwoRA*-forward | This paper | PCR primers | TCAAGTATGAGAGCGAAGCAGC |
| Sequence-based reagent | *cwoRA*-reverse | This paper | PCR primers | TGTCTTATTACGTCTTCCGGTGG |
| Sequence-based reagent | *cwoRB*-forward | This paper | PCR primers | GTATGAGAGCAAGATCCACTTTCC |
| Sequence-based reagent | *cwoRB*-reverse | This paper | PCR primers | GATGATCTCCGTCTTCTCGATAC |
| Sequence-based reagent | *cwoRC*-forward | This paper | PCR primers | GTATGAGAGCCAAGCGACCAC |
| Sequence-based reagent | *cwoRC*-reverse | This paper | PCR primers | CCAAATCCATCTGTCTGCCTC |
| Commercial assay or kit | Direct-zol RNA MiniPrep kit | Zymo Research | Zymo Research: R2050 | |
| Commercial assay or kit | iSCRIPT cDNA synthesis kit | Bio-RAD | Bio-RAD: 1708891 | |
| Commercial assay or kit | iTaq Universal SYBR Green Supermix | Bio-RAD | Bio-RAD: 1725121 | |
| Chemical compound, drug | D-Luciferin, Potassium Salt | Goldbio | Goldbio: LUCK-1G | |
| Chemical compound, drug | TRIzol Reagent | Invitrogen | Thermo Fisher Scientific: 15596026 | |
| Software, algorithm | FaasX software | *Grima et al., 2002* | | http://neuro-psi.cnrs.fr/spip.php?article298&lang=en |
| Software, algorithm | MATLAB (MathWorks) signal-processing toolbox | *Levine et al., 2002* | MATLAB RRID: SCR_001622 | |
| Software, algorithm | MS Excel | | RRID: SCR_016137 | |
| Software, algorithm | GraphPad Prism version 7.0 c for Mac OS X | GraphPad Software, La Jolla, CA USA | RRID: SCR_002798 | www.graphpad.com |

## Fly stocks

Flies were raised on a standard cornmeal/agar medium at 25°C under a 12 hr:12 hr light:dark (LD) cycle. The following *Drosophila* strains were used: $w^{1118}$ – *w; tim-GAL4, UAS-dicer2/CyO (TD2)* (*Dubruille et al., 2009*) – *y w; Pdf-GAL4, UAS-dicer2/CyO (PD2)* (*Dubruille et al., 2009*) – *y w; Tim-GAL4/CyO (TG4)* (*Kaneko et al., 2000*) – *y w; Pdf-GAL4 (PG4)* (*Renn et al., 1999*) – *w;; UAS-dcr2* (*Dietzl et al., 2007*) – *y w;; timHA* (*Rutila et al., 1998*) – *yw; TD2; Pdf-Gal80, Pdf-GAL80* (*Zhang and Emery, 2013*). The following combinations were generated for this study: *y w; TG4; Pdf-GAL80, Pdf-GAL80 – w; tim-GAL4/CyO; UAS-dicer2/TM6B – tim⁰,TG4/CyO; UAS-Dcr2/TM6B – tim⁰, PsiRNAiKK/CyO; timHA/TM6B. TD2, ptim-TIM-LUC* and *TD2, BG-LUC* transgenic flies expressing a *tim-luciferase* and *per-luciferase* fusion gene respectively, combined with the TD2 driver, were used for luciferase experiments. The TIM-LUC fusion is under the control of the *tim* promoter (ca. 5 kb) and 1st intron (*Lamba et al., 2018*), BG-LUC contains per genomic DNA encoding the N-terminal two-thirds of PER and is under the control of the *per* promoter (*Stanewsky et al., 1997*). RNAi lines (names beginning with JF, GL, GLV, HM or HMS) were generated by the Transgenic RNAi Project at Harvard Medical School (Boston, MA) and obtained from the Bloomington Drosophila Stock Center (Indiana University, USA). RNAi lines (names beginning with GD or KK) and control lines (host strain for the KK library containing landing sites for the RNAi transgenes, *VIE-260B*, and *tio* misexpression control strain, *40D-UAS*) were obtained from the Vienna Drosophila Stock Center. *UAS-Psi* flies were kindly provided by D. Rio (*Labourier et al., 2001*).

## Behavioral monitoring and analysis

The locomotor activity of individual male flies (2–5 days old at start of experiment) was monitored in Trikinetics Activity Monitors (Waltham, MA). Flies were entrained to a 12:12 LD cycle for 3–4 days at 25°C (unless indicated) using I-36LL Percival incubators (Percival Scientific, Perry IA). After entrainment, flies were released into DD for five days. Rhythmicity and period length were analyzed using the FaasX software (courtesy of F. Rouyer, Centre National de la Recherche Scientifique, Gif-sur-Yvette, France) (*Grima et al., 2002*). Rhythmicity was defined by the criteria – power $\geq 20$, width $\geq 1.5$ using the $\chi 2$ periodogram analysis. Actograms were generated using a signal-processing toolbox implemented in MATLAB (MathWorks), (*Levine et al., 2002*). For phase-shifting experiments, groups of 16 flies per genotype were entrained to a 12:12 LD cycle for 5–6 days at 25°C exposed to a 5 min pulse of white fluorescent light (1500 lux) at different time points on the last night of the LD cycle. A separate control group of flies was not light-pulsed. Following the light pulse, flies were released in DD for six days. To determine the amplitude of photic phase shifts, data analysis was done in MS Excel using activity data from all flies, including those that were arrhythmic according to periodogram analysis. Activity was averaged within each group, plotted in Excel, and then fitted with a 4 hr moving average. A genotype-blind observer quantified the phase shifts. The peak of activity was found to be the most reliable phase marker for all genotypes. Phase shifts were calculated by subtracting the average peak phase of the light-pulsed group from the average peak phase of non-light pulsed group of flies. Temperature entrainment was performed essentially as described in *Busza et al. (2007)*. Flies were entrained for 4–5 days in LD followed by 11 days in an 8 hr phase advanced temperature cycle. Behavior was analyzed between day 7 and day 10 of the temperature cycle. Actograms were used to ensure that all genotypes had reached – as expected from *Busza et al. (2007)* – a stable phase relationship with the temperature cycle. The phase of the evening peak of activity was determined as described for the phase response curve above. Because, under a LD cycle, the evening peak tend to be truncated by the light off transition, we used the approach described in *Harrisingh et al. (2007)*, which compares the percent of activity between ZT17.5–23.5 that occurs between ZT20.5–23.5 (Morning anticipation phase score), or the percent of activity between ZT5.5–11.5 that occurs between ZT8.5–11.5 (Evening anticipation phase score). If phase is advanced, and activity increases earlier than normal, this percent will decrease.

## Statistical analysis

For the statistical analysis of behavioral and luciferase period length, Student's t-test was used to compare means between two groups, and one-way analysis of variance (ANOVA), coupled to post hoc tests, was used for multiple comparisons. Tukey's post hoc test was used when comparing three or more genotypes and Dunnett's post hoc test was used when comparing two experimental

genotypes to one control. For the statistical analysis of qPCR and the behavioral phase-shifting experiments, two-way ANOVA, coupled to Tukey's post hoc test, was used for multiple comparisons. Statistical analyses were performed using GraphPad Prism version 7.0 c for Mac OS X, GraphPad Software, La Jolla California USA, www.graphpad.com. P values and 95% Confidence Intervals are reported in data source files 'Figure statistics'.

## Luciferase experiments

The luciferase activity of whole male flies on Luciferin (Gold-biotech) containing agar/sucrose medium (170 µl volume, 1% agar, 2% sucrose, 25 mM luciferin), was monitored in Berthold LB960 plate reader (Berthold technologies, USE) in l-36LL Percival incubators with 90% humidity (Percival Scientific, Perry IA). Three flies per well were covered with needle-poked Pattern Adhesive PTFE Sealing Film (Analytical sales and services 961801). The distance between the agar and film was such that the flies were not able to move vertically. Period length was determined from light measurements taken during the first two days of DD. The analysis was limited to this window because TIM-LUC and BG-LUC oscillations severely dampened after the second day of DD. Period was estimated by an exponential dampened cosinor fit using the least squares method in MS Excel (Solver function).

## Real-time quantitative PCR

Total RNA from about 30 or 60 fly heads collected at CT 3, CT9, CT15 and CT21 on the first day of DD were prepared using Trizol (Invitrogen) and Zymo Research Direct-zol RNA MiniPrep kit (R2050) following manufacturer's instructions. 1 µg of total RNA was reverse transcribed using Bio-RAD iSCRIPT cDNA synthesis kit (1708891) following manufacturer's instructions. Real-time PCR analysis was performed in triplicate (three technical replicates per sample) using Bio-RAD iTaq Universal SYBR Green Supermix (1725121) in a Bio-RAD C1000 Touch Thermal Cycler instrument. A standard curve was generated for each primer pair, using RNA extracted from wild-type fly heads, to verify amplification efficiency. Data were normalized to *RpL32* (*Dubruille et al., 2009*) using the $2^{-\Delta\Delta Ct}$ method. Primers used: *RpL32*-forward ATGCTAAGCTGTCGCACAAA; *RpL32*-reverse GTTCGATCCGTAACCGATGT; *psi*-forward GGTGCCTTGAATGGGTGAT; *psi*-reverse CGATTTATCCGGGTCCTCG; *tim-M*-forward TGGGAATCTCGCCCGAAAC; *tim-M*-reverse AGAAGGAGGAGAAGGAGAGAGG; *tim-sc*-forward ACTGTGCGATGACTGGTCTG; *tim-sc*-reverse TGCTTCAAGGAAATCTTCTG; *tim-cold*-forward CCTCCATGAAGTCCTCGTTCG; *tim-cold*-reverse ATTGAGCTGGGACACCAGG; *cwo*-foward TTCCGCTGTCCACCAACTC; *cwo*-reverse CGATTGCTTTGCTTTACCAGCTC; *cwoRA*-forward TCAAGTATGAGAGCGAAGCAGC; *cwoRA*-reverse TGTCTTATTACGTCTTCCGGTGG; *cwoRB*-forward GTATGAGAGCAAGATCCACTTTCC; *cwoRB*-reverse GATGATCTCCGTCTTCTCGATAC; *cwoRC*-forward GTATGAGAGCCAAGCGACCAC; *cwoRC*-reverse CCAAATCCATCTGTCTGCCTC.

## Acknowledgements

We are particularly grateful to Vincent van der Vinne for his help with the analysis of the luciferase recording. We also express our gratitude to Monika Chitre for help with qPCR, Pallavi Lamba for help with PRC, Elaine Chang for help with luciferase recordings, and Diana Bilodeau-Wentworth, Dianne Szydlik, Chunyan Yuan and Vinh Phan for technical assistance. We also thank Dr. Donald Rio as well as the Bloomington and Vienna Drosophila Resource Centers for fly stocks. This work was supported by MIRA award 1R35GM118087 from the National Institute of General Medicine Sciences (NIGMS) to PE, and NIGMS grant 1R01GM125859 to SK.

## Additional information

### Funding

| Funder | Grant reference number | Author |
|---|---|---|
| National Institute of General Medical Sciences | 1R35GM118087 | Patrick Emery |

| National Institute of General Medical Sciences | 1R01GM125859 | Sebastian Kadener |

The funders had no role in study design, data collection and interpretation, or the decision to submit the work for publication.

## Author contributions
Lauren E Foley, Conceptualization, Data curation, Formal analysis, Investigation, Methodology, Writing—original draft, Writing—review and editing; Jinli Ling, Conceptualization, Investigation, Methodology, Writing—review and editing; Radhika Joshi, Investigation, Methodology, Writing—review and editing; Naveh Evantal, Sebastian Kadener, Resources, Writing—review and editing; Patrick Emery, Conceptualization, Formal analysis, Supervision, Funding acquisition, Investigation, Methodology, Writing—original draft, Project administration, Writing—review and editing

## Author ORCIDs
Lauren E Foley (iD) https://orcid.org/0000-0001-7635-7338
Sebastian Kadener (iD) http://orcid.org/0000-0003-0080-5987
Patrick Emery (iD) https://orcid.org/0000-0001-5176-6565

## Decision letter and Author response
Decision letter https://doi.org/10.7554/eLife.50063.SA1
Author response https://doi.org/10.7554/eLife.50063.SA2

# Additional files
## Supplementary files
• Supplementary file 1. RAP Screen Dataset.Circadian behavior analysis for all RNAi lines included in our screen. Period, Power (i. e. rhythm amplitude), and percentage of rhythmic flies are indicated. SD: Standard Deviation. Each lines is crossed to TD2 or PD2, or in some cases to $w^{1118}$.

• Transparent reporting form

## Data availability
All source data are included in this submission.

The following previously published dataset was used:

| Author(s) | Year | Dataset title | Dataset URL | Database and Identifier |
|---|---|---|---|---|
| Wang Q, Tallatero M, Rio D | 2016 | The PSI-U1 snRNP interaction regulates male mating behavior in Drosophila | https://www.ncbi.nlm.nih.gov/geo/query/acc.cgi?acc=GSE79916 | NCBI Gene Expression Omnibus, GSE79916 |

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
