## [Decision Letter]

**Acceptance summary:**

A transcriptional feedback loop comprising of PER/TIMELESS and CLOCK/CYCLE underlies periodic oscillation of the circadian clock and diurnal rhythms. However, the activity of these core components is modulated by post-transcriptional and post-translational processes that remain to be fully understood. This work, by Foley and coauthors, begins to address post-transcriptional control mechanisms by screening *Drosophila* genes encoding RNA-binding and RNA-associated proteins for their role in the control of circadian rhythms. From 364 genes screened a total of 43 candidates (12%) appear to alter period length. The unexpectedly high hit rate suggests that RNA regulation plays a significant role in clock function. The authors then focus on one of these proteins, PSI, and reveal that it encodes a splicing factor that produces timeless splice isoforms described and characterised in a companion paper by Anduaga et al. in this issue of *eLife*. Several observations support a model in which PSI is required to mediate alternative splicing of *tim* isoforms, which in turn determines how the phase of the circadian cycles adapts in response to temperature variation. Observations made here suggest that PSI is required for appropriate *tim* mRNA splicing in both cold and warm conditions, and acts downstream of a primary temperature-sensing mechanism. Additional work is needed to identify how exactly temperature-specific splicing patterns of TIM are determined.

**Decision letter after peer review:**

[Editors’ note: a previous version of this study was rejected after peer review, but the authors submitted for reconsideration. The first decision letter after peer review is shown below.]

Thank you for submitting your work entitled "PSI controls *tim* splicing and

circadian period in *Drosophila*" for consideration by *eLife*. Your article has been reviewed by two peer reviewers, and the evaluation has been overseen by Mani Ramaswami as Reviewing Editor and a Senior Editor. The reviewers have opted to remain anonymous.

Our decision has been reached after consultation between the reviewers. Based on these discussions and the individual reviews below, we regret to inform you that your work will not be considered further for publication in *eLife*.

The reviewers concur that your careful screen of RNA-binding proteins is well done and broadly useful. However, the consensus view is that the behavioural evidence to support the model that PSI mediated splicing of *Tim* contributes to locomotor rhythms is too subtle to be convincing. It also remains possible that thermal adaption affected by *psi* mutations have nothing to do with locomotor rhythms but perhaps to with do other non-rhythmic phenotypes. Thus, the feeling is that despite the interest of the subject and the overall quality of the work, the manuscript falls short of biological insight required for acceptance.

Reviewer #1:

This is an interesting paper focusing on a splicing factor PSI that regulates timeless isoforms. The screening and experimental work is done well with care taken to exclude background effects. One of the disappointing aspects of the study, and this is no fault of the authors, is the lack of any striking disruption of temperature adaptations with the PSI KD. With all these changes in timeless and *cwo* splicing, particularly of the temperature sensitive *tim* isoforms, the locomotor rhythms of the *psi* KD flies in LD cycles is normal without the kind of dramatic effects seen on siestas when per 3' splicing is disturbed. So I was left scratching my head as to what disruption of *psi* does phenotypically – small changes in period, sure, but these are not reflected in the LD profiles of Figure 3—figure supplement 2.

As the period seems to change with *psi* manipulation, I was surprised that the authors did not perform a simple temperature compensation experiment (or did I miss it?). I realize that gal4 is temperature sensitive but the direction of any period change between 18-30° C might have been interesting. Are there compensatory changes of *tim* transcripts in *psi* KD? The accompanying paper hinted at that with the cold-sensitive isoforms.

The loss of rhythmicity in Figure 2E and 2G on *psi* overexpression is striking. Any idea what happens to *tim* isoforms under these conditions? Clearly something in the dorsal neurons is having a major effect on the rhythmicity of PDF neurons. This is interesting as the DNs do not by themselves generate locomotor rhythms. Surprisingly, the Discussion does not mention this most dramatic result of the manuscript.

Reviewer #2:

This is a very interesting paper from the Kadener and Emery labs about a genetic screen for RNA-binding and RNA-associated genes involved in circadian clock control. After screening 364 genes, they found a total of 43 candidates (12%) that alter period length. Compared to standard genetic screens, this is a very high hit rate, indicating that RNA regulation plays a significant role in clock regulation. If correct, this is an important discovery revealing yet another regulatory level of the already very complex circadian clock mechanism.

The paper is well written and experiments conducted are of high quality and according to the standard in the field. The authors focus on one gene identified in their screen (*psi*), which they show causes period shortening when down-regulated and arrhythmicity when overexpressed in the entire clock circuit. Effects with more narrow down regulation (*Pdf-gal4*) are much less pronounced and overall it remains unclear in which parts of the circuit *Psi* plays a role. Moreover important controls are lacking in the behavior experiments (outlined below). Because the effects on period are rather mild, these controls are important. Also in light of the fact that *psi* has no measurable effect on the light PRC or normal LD entrainment it seems important to really nail the effect on period. Molecularly it seems clear that *psi* affects *tim*-splicing, but the mechanism remains elusive and it is not clear how *psi* can promote intron retention in certain *tim* transcripts at cooler temperature and that in other *tim* transcripts at warmer temperatures.

1) The period shortenings after *Psi*-knockdown are relatively mild (0.2-1.2 hr depending on the driver used) and the same applies for the lengthening observed after *psi*-overexpression (0.1 – 0.7 hr) (Figure 2 and Table 3). Given that these period changes are key to the message of the paper (*psi* controls circadian period), I think it is important to perform a statistical analysis comparing the various genotypes. Also, some key controls are missing: the *psi* RNAi lines have not been tested without a driver. Data are shown for the total of RNAi-lines from a given collection over + in Figure 1A, but due to the large variability between the different collections with regard to period length, the values for the individual RNAi lines 'over +' must be given. I also checked the source file (Dataset 1) and found some data, which I think shows the RNAi/+ data. From these data it looks the effect of the *psi* knockdown is even smaller (*psiRNAi-kk*/+: 24.0, *TD2>kk*: 23.6, *PD2>kk*:24.7 and *psiRNAi-GD*/+ 24.2, *TD2>GD*: 23.8, *PD2>GD*: 24.2). So no effect with the *Pdf* driver on period at all, and a small 0.4 hr effect with *TD2*. Moreover, no effects with 2 other RNAi lines shown in this Data set with either *tim* or *Pdf* drivers. I am not sure if the authors have more control data and if the Data set table represents only the actual screening data, but clearly additional controls are required. This is an important point, because no other behavioral phenotypes after *psi* knockdown could be observed (LD behavior at different temperatures and light-PRC, Figure 3—figure supplements 2 and 3).

2) The authors see stronger effects with *tim-gal4* (both knock-down and overexpression of *psi*) compared to *Pdf-gal4*. They aim to distinguish between potentially stronger *tim-gal4* expression in the s-LNv, or a period-determining function of non-s-LNv neurons by applying a *tim-gal4/Pdf-gal80* combination, which should eliminate *gal4* from the s-LNv but not all the other clock cells. Unfortunately, this experiment did not allow for a clear distinction, as the period shortening after knockdown was in between that of *tim* and *Pdf* drivers alone. Overexpression with the *tim-gal4/Pdf-gal80* combination recapitulated the high percentage of arhythmicity seen with tim alone, but not the period-lengthening, whereas overexpression with *Pdf-gal4* only had no effect. Overall the results suggest a role for non-s-LNv neurons in setting period length and controlling rhythmicity, but I think this should be repeated with other, non-s-LNv drivers to get a clearer picture (e.g., it cannot be ruled out that *Pdf-gal80* is completely blocking gal4 in the s-LNv). I suggest using the splitE cell-gal4 and drivers specific for the DN1 to solve this issue.

3) *psi* overexpression with *tim-gal4* and *tim-gal4/Pdf-gal80* produces very high levels of arrhythmicity and the few rhythmic flies (n=5 and n=4, respectively) have low average power values. Considering the low number of rhythmic flies and the low power values, I find it problematical to claim that overexpression with *tim-gal4* causes period-lengthening. Could the authors show actograms that clearly show the longer period in these flies compared to the *tim-gal4/Pdf-gal80* flies. Lacking additional support (such as convincing actograms and/or higher n's), I think it is not OK to conclude that the overexpression results are in line with the knockdown results (which are problematical in itself, see major point 1 above).

4) The peripheral clock data in Figure 3 do not look very convincing, due to the poor rhythmicity of the luciferase oscillations in DD. As correctly mentioned in the Materials and methods part, it is expected and was reported previously that rhythms of these reporters dampen rapidly in DD. But I am not sure if it is valid to calculate period values from 48 hr only, particularly if one the 2nd day the oscillations are very low amplitude. Perhaps it would be better to look in LD (where the reporters cycle with high-amplitude) to see if the rhythms in *psi* knock down flies are slightly phase-advanced? Or use dissected peripheral tissues in hope of stronger rhythms in DD (whole body rhythms could be further dampened due to internal desynchronization between tissues).

5) Figure 5 and accompanying Results text. I think this is the key figure about the molecular effects of reducing *psi* function (the effects on *cwo* are relatively weak and much less convincing). The data in Figure 5 clearly show that *tim* introns that are usually retained at cold temperatures (due to no splicing at the usual splice sites, I assume) and lead to higher mRNA levels of these particular transcripts, are also elevated at 25°C when *psi* is being knocked down. So it seems that *psi* is responsible for splicing out these introns at warmer temperatures. In contrast, an intron normally retained at high temperatures (29°C) and resulting in high *tim-M* transcript levels is spliced out in *psi* knockdown heads at 25°C, leading to lower *tim-M* levels both at 25°C and 29°C. This suggests that for this transcript, *psi* seems to promote intron retention (so to reduce splicing) at warmer temperatures. How can this be explained by a common molecular mechanism? Also, it doesn't help that the authors refer to an accompanying paper (not available for this reviewer) for the nature of the different *tim* transcripts and splicing events. Without a map explaining these mainly totally novel alternative *tim* transcripts it is impossible to follow what is going on. So, a map seems mandatory, also because the reader should not be forced to swap to a different paper in order to understand the current one.

6) Figure 6 supports the idea that *psi* regulates period by *tim* splicing. I am bit confused by the genotype description used in the figure and Table 3. In the other parts of the paper *tim-gal4 UAS-dicer2* was abbreviated as *TD2*, but here '*TG4*/+;*UAS-Dcr2*/+ was used. Are these the same flies or constructs on different chromosomes. Please explain. Also, as in Figure 2, the controls for RNAi/+ are missing.

[Editors’ note: what now follows is the decision letter after the authors submitted for further consideration.]

Thank you for submitting your article "*Drosophila* PSI controls circadian period and the phase of circadian behavior under temperature cycle via *tim* splicing" for consideration by *eLife*. Your article has been reviewed by two peer reviewers, and the evaluation has been overseen by Mani Ramaswami as Reviewing Editor and Ronald Calabrese as the Senior Editor. The reviewers have opted to remain anonymous.

The reviewers have discussed the reviews with one another and the Reviewing Editor has drafted this decision to help you prepare a revised submission.

Summary:

This is a resubmitted version of a previously declined article that has been substantially improved. The work described begins with a genetic screen for RNA-binding and RNA-associated genes involved in circadian clock control. Of 364 genes screened, a total of 43 candidates (12%) appear to alter period length. The unexpectedly high hit rate compared to standard genetic screens is significant and interesting, suggesting that RNA regulation plays a significant role in clock regulation. The authors go on to focus on one of these, PSI, a splicing factor that regulates timeless isoforms. Various observations support a model in which PSI controlled temperature-dependent-splicing regulates the circadian period and determines how the phase of the circadian cycles adapts in response to temperature cycles. However, there remain several ambiguities, missing data, controls and issues that need to be addressed.

Essential revisions:

1) Figure 2C to H. At 30°C there are clear period-shortening effects (panel 2C). But from Table 3 and it's difficult for the reader to extract from these numbers whether the UAS-RNAi, and UAS-Psi constructs are significantly different from the experimentals at 25°C. Where are the UAS-Psi RNAi and UAS-Psi overexpression controls for 25°C work? Without these it's difficult to assess how valid are the period changes at 25°C. These should be included. Some of the statements being presented from Figure 2 are possibly incorrect e.g. panel 2D. The authors need to present the 25°C UAS/+ results graphically in Figure 2 and perform the appropriate statistical comparisons. In fact the UAS *psi*/+ result would actually strengthen their case for period lengthening for panels F and H.

2) Figure 3A and B isn't so convincing visually because there is only one cycle, even though the cosinor method used is appropriate. Might this be relegated to a supplementary figure as it adds very little to the manuscript?

3) Further in Figure 3: It is still very difficult to see the period shortening in the bioluminescence traces (B, D). In particular it is difficult to distinguish between the black circles and black squares (RNAi/+ and test flies, respectively). It would help to at least use colored symbols to help distinguishing the different genotypes.

4) Figure 4 does not add much to this manuscript, where two *cwo* splice forms appear to be reduced and one is increased. The results are described briefly then immediately dropped – this could be relegated to a supplementary figure.

5) The *tim* splicing results are very confusing and need to be clarified. The cold *tim* isoforms (*tim-cold* and *tim-sc*) are elevated at warm temperatures in the *psi* mutant so normally *psi* would enhance splicing at cold temperatures. The warmer *tim* isoform (*tim-M*) is reduced at 25°C in the *psi* mutant so we'd presume that *psi* normally enhances this isoform at warm temperatures. Therefore *psi* normally has opposite effects on the *tim* isoforms, so a general temperature-sensitive *psi* mechanism appears to be excluded.

6) Figure 4—figure supplement 1 the UAS RNAi controls are not shown here. Are they temperature compensated? The Td2 control is not temperature compensated and has a relatively large change in period at hot temperature.

7) 'Collectively, these results indicate that PSI shifts the balance toward a warm temperature *tim* RNA isoform profile at an intermediate temperature (25°C).' This sentence makes no sense – no general mechanism can emerge as PSI appears to have opposite effects on the cold/warm *tim* isoforms. It enhances the warm forms and reduces the cold forms. Do the authors mean wild-type PSI or the *psi*-mutant? Please clarify.

---

## [Author Response]

[Editors’ note: the author responses to the first round of peer review follow.]

We would like to thank the two reviewers for their insightful comments. We believe that by addressing them with additional experiments and improved discussion, we have very significantly improved our manuscript. The most important additions are the followings. First, we have discovered that PSI downregulation advances the phase of circadian behavior under temperature cycles. This phenotype is remarkably strong and reveal a specific role for PSI in circadian thermal response. Second, we have also significantly strengthened the data supporting a role for PSI in period control.

Reviewer #1:This is an interesting paper focusing on a splicing factor PSI that regulates timeless isoforms. The screening and experimental work is done well with care taken to exclude background effects. One of the disappointing aspects of the study, and this is no fault of the authors, is the lack of any striking disruption of temperature adaptations with the PSI KD. With all these changes in timeless and cwo splicing, particularly of the temperature sensitive tim isoforms, the locomotor rhythms of the psi KD flies in LD cycles is normal without the kind of dramatic effects seen on siestas when per 3' splicing is disturbed. So I was left scratching my head as to what disruption of psi does phenotypically – small changes in period, sure, but these are not reflected in the LD profiles of Figure 3—figure supplement 2.As the period seems to change with psi manipulation, I was surprised that the authors did not perform a simple temperature compensation experiment (or did I miss it?). I realize that gal4 is temperature sensitive but the direction of any period change between 18-30°C might have been interesting. Are there compensatory changes of tim transcripts in psi KD? The accompanying paper hinted at that with the cold-sensitive isoforms.

We thank the reviewer for his interest in our work. We did perform a temperature compensation experiment but did not observe a phenotype. We have added these results to the manuscript. However, we have now also performed temperature entrainment experiments and observed a striking phenotype.The phase of circadian behavior is advanced by several hours under a temperature cycle. It is however not advanced under a light/dark cycle. Moreover, in flies that cannot splice *tim* in a temperature dependent manner, the phase of circadian behavior is not advanced. Therefore, PSI specifically regulates the phase of circadian behavior in the presence of a temperature cycle, through regulation of *tim* splicing.

The loss of rhythmicity in Figure 2E and 2G on psi overexpression is striking. Any idea what happens to tim isoforms under these conditions? Clearly something in the dorsal neurons is having a major effect on the rhythmicity of PDF neurons. This is interesting as the DNs do not by themselves generate locomotor rhythms. Surprisingly, the Discussion does not mention this most dramatic result of the manuscript.

Reviewer 2 also showed interest in this phenotype, which we tried to attribute to a specific group of circadian neurons. Unfortunately, the results are not straightforward, and suggest that arrhythmicity is the sum of PSI downregulation in multiple groups of circadian neurons. Also, as we discussed in the original manuscript, arrhythmicity is more difficult to interpret than a period phenotype. We would thus prefer not to include these data in the present manuscript, particularly since we have now identified a striking temperature-dependent phenotype. However, if the editors and reviewers concur that we should add these data, we will provide them.

Reviewer #2:[…] The paper is well written and experiments conducted are of high quality and according to the standard in the field. The authors focus on one gene identified in their screen (psi), which they show causes period shortening when down-regulated and arrhythmicity when overexpressed in the entire clock circuit. Effects with more narrow down regulation (Pdf-gal4) are much less pronounced and overall it remains unclear in which parts of the circuit Psi plays a role. Moreover important controls are lacking in the behavior experiments (outlined below). Because the effects on period are rather mild, these controls are important. Also in light of the fact that psi has no measurable effect on the light PRC or normal LD entrainment it seems important to really nail the effect on period. Molecularly it seems clear that psi affects tim-splicing, but the mechanism remains elusive and it is not clear how psi can promote intron retention in certain tim transcripts at cooler temperature and that in other tim transcripts at warmer temperatures.1) The period shortenings after Psi-knockdown are relatively mild (0.2-1.2 hr depending on the driver used) and the same applies for the lengthening observed after psi-overexpression (0.1 – 0.7 hr) (Figure 2 and Table 3). Given that these period changes are key to the message of the paper (psi controls circadian period), I think it is important to perform a statistical analysis comparing the various genotypes. Also, some key controls are missing: the psi RNAi lines have not been tested without a driver. Data are shown for the total of RNAi-lines from a given collection over + in Figure 1A, but due to the large variability between the different collections with regard to period length, the values for the individual RNAi lines 'over +' must be given. I also checked the source file (Dataset 1) and found some data, which I think shows the RNAi/+ data. From these data it looks the effect of the psi knockdown is even smaller (psiRNAi-kk/+: 24.0, TD2>kk: 23.6, PD2>kk:24.7 and psiRNAi-GD/+ 24.2, TD2>GD: 23.8, PD2>GD: 24.2). So no effect with the Pdf driver on period at all, and a small 0.4 hr effect with TD2. Moreover, no effects with 2 other RNAi lines shown in this Data set with either tim or Pdf drivers. I am not sure if the authors have more control data and if the Data set table represents only the actual screening data, but clearly additional controls are required. This is an important point, because no other behavioral phenotypes after psi knockdown could be observed (LD behavior at different temperatures and light-PRC, Figure 3—figure supplements 2 and 3).

Indeed, at first sight, it might appear that the PSI RNAi phenotypes are quite modest. Because the *tim-GAL4* and *pdf-GAL4* drivers cause a ca. 1hr period lengthening on their own (this has been observed since these drivers were first reported in the late 90s by the Hall lab), the only meaningful period comparison is between driver/+ flies and drivers/RNAi flies, which is what we showed on the figures. Importantly PSI came out of a screen of over 600 RNAi lines and was one of the rare genes to show a short period phenotype when downregulated, so it is clear that a 1.5 hour period shortening compared to driver control is highly significant, and specific. What the RNAi/+ control in dataset1 show is that the RNAi transgenes, on their own, do not shorten period. This is now explained more thoroughly in the manuscript. We would like to mention that in the late stages of preparing our original manuscript, a portion of the text that contained much of these explanations was inadvertently deleted. We apologize for this oversight, which probably contributed to the concerns of the reviewer.

We were however able to further support our claim that PSI regulates period. When we monitored circadian behavior at 30°C instead of 25°C, we observed that the circadian period of the *timGAL4*/+ controls was not significantly different from the RNAi/+ controls. We could therefore meaningfully compare both RNAi/+ and *timGAL4*/+ controls to the experimental flies. The results are clear, period is shorter in the experimental flies than in the controls. This was added to Figure 2C.

We also considerably increased the number of PSI overexpressing flies tested, and confirmed that period is indeed long in these flies.

Thus, we have clearly established that PSI levels are critically important for the period length of circadian behavior.

2) The authors see stronger effects with tim-gal4 (both knock-down and overexpression of psi) compared to Pdf-gal4. They aim to distinguish between potentially stronger tim-gal4 expression in the s-LNv, or a period-determining function of non-s-LNv neurons by applying a tim-gal4/Pdf-gal80 combination, which should eliminate gal4 from the s-LNv but not all the other clock cells. Unfortunately, this experiment did not allow for a clear distinction, as the period shortening after knockdown was in between that of tim and Pdf drivers alone. Overexpression with the tim-gal4/Pdf-gal80 combination recapitulated the high percentage of arhythmicity seen with tim alone, but not the period-lengthening, whereas overexpression with Pdf-gal4 only had no effect. Overall the results suggest a role for non-s-LNv neurons in setting period length and controlling rhythmicity, but I think this should be repeated with other, non-s-LNv drivers to get a clearer picture (e.g., it cannot be ruled out that Pdf-gal80 is completely blocking gal4 in the s-LNv). I suggest using the splitE cell-gal4 and drivers specific for the DN1 to solve this issue.

As described above in response to one of Reviewer #2’s comments, we tried to map arrhythmicity to specific neurons, but the results are not straightforward, and an arrhythmic phenotype is more difficult to interpret than a change in period We would thus prefer to leave these data out of the manuscript to keep the focus on the most important and solid results, but if the editors and reviewers consider it necessary for us to add these data, we will do so.

3) psi overexpression with tim-gal4 and tim-gal4/Pdf-gal80 produces very high levels of arrhythmicity and the few rhythmic flies (n=5 and n=4, respectively) have low average power values. Considering the low number of rhythmic flies and the low power values, I find it problematical to claim that overexpression with tim-gal4 causes period-lengthening. Could the authors show actograms that clearly show the longer period in these flies compared to the tim-gal4/Pdf-gal80 flies. Lacking additional support (such as convincing actograms and/or higher n's), I think it is not OK to conclude that the overexpression results are in line with the knockdown results (which are problematical in itself, see major point 1 above).

We agree that the number of rhythmic flies was low. We have repeated these experiments and considerably increased the Ns. We confirmed our initial results.

4) The peripheral clock data in Figure 3 do not look very convincing, due to the poor rhythmicity of the luciferase oscillations in DD. As correctly mentioned in the Materials and methods part, it is expected and was reported previously that rhythms of these reporters dampen rapidly in DD. But I am not sure if it is valid to calculate period values from 48 hr only, particularly if one the 2nd day the oscillations are very low amplitude. Perhaps it would be better to look in LD (where the reporters cycle with high-amplitude) to see if the rhythms in psi knock down flies are slightly phase-advanced? Or use dissected peripheral tissues in hope of stronger rhythms in DD (whole body rhythms could be further dampened due to internal desynchronization between tissues).

We have not been very successful with dissected organs in our lab in DD. In LD, the effect would be expected to be minimal based on behavior (see Figure 3—figure supplement 2), and we think it would be unlikely to observe a significant effect. However, we are confident that the results presented are solid. Phase changes are clearly visible on the graphs, and period was reproducibly shorter in experimental flies in all experiments, even though for one reporter we were slightly above a P value of 0.05. In addition, the period shortening observed with luciferase is very similar to that observed with locomotor behavior.

5) Figure 5 and accompanying Results text. I think this is the key figure about the molecular effects of reducing psi function (the effects on cwo are relatively weak and much less convincing). The data in Figure 5 clearly show that tim introns that are usually retained at cold temperatures (due to no splicing at the usual splice sites, I assume) and lead to higher mRNA levels of these particular transcripts, are also elevated at 25°C when psi is being knocked down. So it seems that psi is responsible for splicing out these introns at warmer temperatures. In contrast, an intron normally retained at high temperatures (29°C) and resulting in high tim-M transcript levels is spliced out in psi knockdown heads at 25°C, leading to lower tim-M levels both at 25°C and 29°C. This suggests that for this transcript, psi seems to promote intron retention (so to reduce splicing) at warmer temperatures. How can this be explained by a common molecular mechanism? Also, it doesn't help that the authors refer to an accompanying paper (not available for this reviewer) for the nature of the different tim transcripts and splicing events. Without a map explaining these mainly totally novel alternative tim transcripts it is impossible to follow what is going on. So, a map seems mandatory, also because the reader should not be forced to swap to a different paper in order to understand the current one.

We regret that the reviewer could not access the accompanying paper. However, we agree that a map of *tim* splicing should be added to our manuscript and we have done so. In terms of mechanisms for differential effects on different splicing events, it is possible that one splicing event regulates the probability of the other. We are now mentioning this possibility in the Discussion.

6) Figure 6 supports the idea that psi regulates period by tim splicing. I am bit confused by the genotype description used in the figure and Table 3. In the other parts of the paper tim-gal4 UAS-dicer2 was abbreviated as TD2, but here 'TG4/+;UAS-Dcr2/+ was used. Are these the same flies or constructs on different chromosomes. Please explain. Also, as in Figure 2, the controls for RNAi/+ are missing.

Yes, we used a UAS-Dcr2 transgene on a different chromosome, because we needed to be in a *tim^0^*background for the experiments shown on Figure 7. *tim* is on the second chromosome, so to build the flies it was easier to use a 3^rd^ chromosome *UAS-Dcr2* insertion. This is more clearly explained in the figure legend.

[Editors' note: the author responses to the re-review follow.]

Essential revisions:1) Figure 2C to H. At 30°C there are clear period-shortening effects (panel 2C). But from Table 3 and it's difficult for the reader to extract from these numbers whether the UAS-RNAi, and UAS-Psi constructs are significantly different from the experimentals at 25°C. Where are the UAS-Psi RNAi and UAS-Psi overexpression controls for 25°C work? Without these it's difficult to assess how valid are the period changes at 25°C. These should be included. Some of the statements being presented from Figure 2 are possibly incorrect e.g. panel 2D. The authors need to present the 25°C UAS/+ results graphically in Figure 2 and perform the appropriate statistical comparisons. In fact the UAS psi/+ result would actually strengthen their case for period lengthening for panels F and H.

We have added the *PSI RNAi/+* controls to Figure 2A. They show that the RNAi lines, on their own, do not shorten period, compared to wild-type flies. In addition, the experimental flies are statistically shorter than these controls. However, the difference is smaller than against *tim-GAL4, Uas-dicer2 (TD2)/+* controls, because *tim-GAL4* causes a well-known ca. 0.8 hr period lengthening at 25°C. Because this period lengthening is dominant, the key comparison to understand the impact of PSI downregulation on circadian behavior is *TD2/+* vs. *TD2/Psi RNAi*. This is the case for the PD2 combination as well, because *pdf-GAL4* similarly lengthens period in a dominant manner. The effect on period is weaker when RNAi expression is restricted to PDF cells with *PD2*, or to non-PDF cells with *TD2/-pdf-GAL8*0. As a result, while the experimental flies are statistically significantly shorter than *PD2/+* or *TD2/+; pdfGAL80/+* control flies, they are not shorter than *PSI-RNAI/+* controls. This does not invalidate our conclusions that both PDF and non-PDF cells are implicated in the PSI phenotype. The presence of the RNAi transgenes, on their own, has no effect on period (new Figure 2A), and the comparison that has to be done, to evaluate the contribution of *PSI-RNAi* expression, is experimental flies vs. the driver controls (*PD2/+* or *TD2/+; pdfGAL80*/+) because of their dominant effect on period. To make sure that the figure is not confusing to the reader, we only showed the key genotype comparisons in Figure 2B and C, but in the figure legend and the main text we made every effort to make clear to the reader why these are the key comparisons, and that the RNAi on their own do not impact period.

We have now also included the *UAS-PSI/+* controls in Figure 2F-H. As expected, experimental flies are statistically longer than these control flies, but again, the key comparison is actually experimental flies vs. GAL4 driver controls.

2) Figure 3A and B isn't so convincing visually because there is only one cycle, even though the cosinor method used is appropriate. Might this be relegated to a supplementary figure as it adds very little to the manuscript?

As suggested, we have now moved this figure to the supplementary figures (Figure 2—figure supplement 2A, B). Please not however, that there are two cycles on the figures, not just one.

3) Further in Figure 3: It is still very difficult to see the period shortening in the bioluminescence traces (B, D). In particular it is difficult to distinguish between the black circles and black squares (RNAi/+ and test flies, respectively). It would help to at least use colored symbols to help distinguishing the different genotypes.

Thank you for this suggestion, we have improved the presentation of this figure.

4) Figure 4 does not add much to this manuscript, where two cwo splice forms appear to be reduced and one is increased. The results are described briefly then immediately dropped – this could be relegated to a supplementary figure.

As suggested, we have moved this figure to the supplementary figures (Figure 3—figure supplement 1).

5) The tim splicing results are very confusing and need to be clarified. The cold tim isoforms (tim-cold and tim sc) are elevated at warm temperatures in the psi mutant so normally psi would enhance splicing at cold temperatures. The warmer tim isoform (tim-M) is reduced at 25°C in the psi mutant so we'd presume that psi normally enhances this isoform at warm temperatures. Therefore psi normally has opposite effects on the tim isoforms, so a general temperature-sensitive psi mechanism appears to be excluded.

We have made additional efforts in the text (Results and Discussion) to make clear the impact of PSI on *tim* splicing. Briefly, PSI promotes splicing favored at warm temperature, while inhibiting those favored at cold temperature, and it does this at any temperature. As we discussed, while PSI determines the ratio of *tim* isoforms, their temperature sensitivity might be encoded through splice site strength, as for *per* 3’UTR intron retention event. In the Discussion, we proposed that splicing events might be co-regulated, and that therefore favoring one can, at the same time, indirectly influence the probability of another (excerpt from the previous Discussion: “Another interesting question is how PSI affects differentially specific splice *tim* isoforms. One possibility is that the execution of a specific *tim* splicing event influences that of another”). We have expended this point of Discussion to make it clearer. We believe that with this addition and a few minor text corrections in the relevant Results section, the impact of PSI on *tim* splicing should be entirely clear to the reader.

6) Figure 4—figure supplement 1 the UAS RNAi controls are not shown here. Are they temperature compensated? The Td2 control is not temperature compensated and has a relatively large change in period at hot temperature.

They are temperature compensated. We had left them out because of the dominant impact of *TD2*, but we have now added them.

7) 'Collectively, these results indicate that PSI shifts the balance toward a warm temperature tim RNA isoform profile at an intermediate temperature (25°C).' This sentence makes no sense – no general mechanism can emerge as PSI appears to have opposite effects on the cold/warm tim isoforms. It enhances the warm forms and reduces the cold forms. Do the authors mean wild-type PSI or the psi-mutant? Please clarify.

Please see our response above on the possibility that a splicing event influences another. Also, we have added “in wild-type flies”. We hope that this sentence now makes sense.